# Composite Slice Transformer: An Efficient Transformer with Composition of Multi-Scale Multi-Range Attentions

**Mingu Lee**[1]    **Saurabh Pitre**[*]  **Tianyu Jiang**[1]    **Pierre-David Letourneau**[1]
**Matthew J. Morse**[1]    **Kanghwan Jang**[1]    **Joseph Soriaga**[1]
**Parham Noorzad**[2]    **Hsin-Pai Cheng**[1]    **Christopher Lott**[1]

[1] Qualcomm AI Research[†]   [2] Qualcomm Technologies Inc.

{mingul, clott}@qti.qualcomm.com

## Abstract

Since the introduction of Transformers, researchers have tackled the expensive quadratic complexity of the attention mechanism. While significant complexity improvements have been achieved, they often come at the cost of reduced accuracy. In this paper, we propose Composite Slice Transformer (**CST**), a Transformer-based network equipped with a composition of multi-scale multi-range attentions, boosting both efficiency and modeling capability. After stacking fixed-length slices of the input sequence, each layer in **CST** performs a pair of fine-and-coarse-grained attentions with short and long ranges in a sequential manner, coupled with a volatile instant positional embedding. In addition to significantly reduced $O(NL+N^2/L^2)$ complexity for sequence length $N$ and slice length $L$, **CST** achieves superior performance on a variety of tasks. We show that **CST** surpasses recently published efficient Transformers on the Long Range Arena benchmark, demonstrating the bidirectional long-range dependency modeling capability of our model with a comparable complexity. It also outperforms the standard Transformer by a margin of 6.9% in average accuracy across the five classification tasks. On the word-level WikiText-103 autoregressive language modeling task with various sequence lengths, and the masked language modeling followed by GLUE benchamrk, **CST** outperforms most other efficient Transformers while being competitive against the Transformer.

## 1 Introduction

Transformers (Vaswani et al., 2017) are one of the most important recent advances in artificial intelligence. Since they can be combined in a straightforward fashion with advanced training methods and auxiliary modules, Transformers have proven extremely effective as a versatile backbone architecture for achieving state-of-the-art performance in many domains such as natural language processing (Devlin et al., 2018; Yang et al., 2019; Brown et al., 2020; Raffel et al., 2020; Sanh et al., 2022), vision processing (Dosovitskiy et al., 2020; Liu et al., 2021b; Radford et al., 2021), visual language modeling (Alayrac et al., 2022), speech recognition (Dong et al., 2018; Gulati et al., 2020; Shi et al., 2021), and reinforcement learning (Chen et al., 2021b; Janner et al., 2021).

Despite this versatility, Transformers possess an expensive memory and computational complexity of $O(N^2)$ with respect to input length $N$ in the multi-head self-attention computation. As a result, Transformers are often not applied to long sequence data. Since the introduction of Transformers in (Vaswani et al., 2017), recent work has focused on improving Transformer complexity through various techniques, achieving efficiency gains in complexity and memory requirements (Tay et al., 2020c), with several models attaining $O(N)$ complexity (Wang et al., 2020; Katharopoulos et al.,

---

[*]Work was completed during employment at Qualcomm Technologies, Inc.

[†]Qualcomm AI Research is an initiative of Qualcomm Technologies, Inc

2020; Ma et al., 2021). Unfortunately, these efficiencies come at the cost of reduced accuracy and often lack the ability to model fine-grained token dependencies, limiting the application of these recent improvements to a wider range of practical problems. Recent studies (Zhu et al., 2021; Ren et al., 2021; Nguyen et al., 2021; Zhu & Soricut, 2021; Han et al., 2021) show that combining efficient techniques for global sequence modeling with fine-grained limited range attention can improve accuracy, maintain low complexity, and enable longer context windows, allowing such models to outperform the standard, full-resolution Transformer on certain tasks. However, optimally combining sequence modeling at different levels of granularity remains an open problem.

To this end, we propose the *Composite Slice Transformer* (**CST**), an efficient Transformer-based network architecture, consisting of a composition of attentions applied to a stacked slice representation of the input sequence at different scales, coupled with a multi-scale volatile instant positional embedding. For fixed slice length $L$, **CST** has a complexity of $O(NL + N^2/L^2)$, which is comparable or more efficient to linear complexity in many practical settings (Section A.4). Since slicing restricts fine-grained token interaction across boundaries, **CST** also leverages an extended local attention, preventing context fragmentation (Dai et al., 2019b) and enabling seamless sequence modeling. These improvements allow **CST** to outperform the standard Transformer on several benchmark tasks. Similar to (Dosovitskiy et al., 2020), **CST** abstracts the input sequence into another with fewer tokens to compute a low-resolution attention with longer range and a fixed-size segment-wise embedding. With a segment length $L$, the reduced sequence has length $N/L$; evaluating attention on this reduced sequence has complexity $O(N^2/L^2)$. With an appropriately chosen $L$, **CST** can achieve a significant complexity reduction without a loss of performance. Along with the segment-level attention, **CST** also leverages a full-resolution local attention to form a *multi-scale multi-range attention* (MSMRA) through a composition with the global attention, improving model expressiveness (Section A.5), unlike (Zhu & Soricut, 2021; Zhu et al., 2021; Nguyen et al., 2021; Ren et al., 2021; Han et al., 2021).

**CST** extends the ideas of position-infused attention (Press et al., 2021) and applies them to MSMRA, which we refer to as multi-scale volatile instant positional embedding (MS-VIPE). In addition to its effectiveness as a positional embedding, MS-VIPE provides further parameter efficiency by only requiring storage of reduced lengths of local ($L$) and global ($N/L$) attentions instead of the full sequence length $N$. We evaluate our model on bidirectional long-range dependency modeling tasks, an autoregressive language modeling task, and natural language understanding tasks. In our experiments (Section 5), **CST** achieves state-of-the-art performance among all Transformer-based models (including the standard Transformer and recently proposed efficient Transformers) and demonstrates strong performance on a wide variety of tasks.

The paper is organized as follows. In Section 2 and Section 3, we discuss recent efficient Transformer developments, discuss strengths and weaknesses, and outline the inspiration for **CST**. In Section 4, we present our model, Composite Slice Transformer, and describe its features. In Section 5, we provide our experimental results on the Long Range Arena, WikiText-103 autoregressive language modeling, and the GLUE benchmarks, demonstrating the efficiency and versatility of **CST**. In Section 6, we conclude with a summary of our work.

## 2 RELATED WORK

Tay et al. (2020c) provides a comprehensive study of proposed approaches to accelerate the attention mechanism of Vaswani et al. (2017). The key differentiating factor among the approaches presented is the modeling assumptions used to approximate the original attention map. For example, common assumptions include sparsity (Child et al., 2019; Ho et al., 2019; Beltagy et al., 2020; Ainslie et al., 2020; Zaheer et al., 2020; Tay et al., 2020a; Kitaev et al., 2019) and low-rankness (Wang et al., 2020; Choromanski et al., 2020; Katharopoulos et al., 2020). Chen et al. (2021a) combines these two assumptions. Other architectures leverage additional memory for compressing the global context (Lee et al., 2019; Ma et al., 2021; Jaegle et al., 2021). In order to capture fine-grained token interactions that might be missing in the abstractive attention, (Zhu & Soricut, 2021; Zhu et al., 2021; Nguyen et al., 2021; Ren et al., 2021) use a process akin to leverage a full-resolution local attention to form a *multi-scale multi-range attention* (MSMRA). These approaches, however, do not compose the local full-resolution attention and global reduced-resolution attentions in a series. Absent from this literature, serial composition and positional embedding would improve accuracy while preserving the efficiency (see the following sections for how we address this enhancement). More recently,

there also have been proposed sequence models with structures or strong inductive biases achieving significant improvements in benchmarks such as (Gu et al., 2021; Mehta et al., 2022; Ma et al., 2022; Li et al., 2022). (Press et al., 2021) propose the position-infused attention (PIA), which is an attention module with a layer-wise positional embedding applied only to queries and keys, to address token representation reusability issue in increasing window attention due to positional information draft, while avoiding expensive computation of relative position encoding (Dai et al., 2019b). Another study of adding positional embedding to values is conducted in (Tsai et al., 2019), concluding that positional embedding on value does not lead to performance improvement. We extend these ideas in **CST** and apply in MSMRA.

## 3 PRELIMINARIES

### 3.1 TRANSFORMERS AND MULTI-HEAD SELF-ATTENTION

A Transformer layer consists of a multi-head self-attention sublayer followed by a feed-forward network (Vaswani et al., 2017) with an optional cross-attention sublayer when used as a decoder. The multi-head self-attention is defined as the concatenation of the self-attention output of all attention heads:

$$\mathbf{Y} = [\mathbf{Y}_0, \mathbf{Y}_1, ..., \mathbf{Y}_{H-1}]_2, \tag{1}$$

where $[\cdot]_r$ denotes concatenation in the $r^{\text{th}}$ dimension, and each of the outputs $\mathbf{Y}_h \in \mathbb{R}^{N \times d_h}$ is a scaled dot-product attention computed from the input $\mathbf{X} \in \mathbb{R}^{N \times D}$ as

$$\mathbf{Y}_h = \text{softmax}\left(\frac{\mathbf{Q}_h \mathbf{K}_h^\top}{\sqrt{d_h}}\right) \mathbf{V}_h = \mathbf{A} \mathbf{V}_h. \tag{2}$$

In Eq. 2, $\mathbf{Q}_h = \mathbf{X} \mathbf{W}_{q,h}$, $\mathbf{K}_h = \mathbf{X} \mathbf{W}_{k,h}$, and $\mathbf{V}_h = \mathbf{X} \mathbf{W}_{v,h}$ are queries, keys and values, respectively, expressed as linear transformations of the input $\mathbf{X}$ by $\mathbf{W}_{\cdot,h} \in \mathbb{R}^{D \times d_h}$. We assume the queries, keys, and values have the same hidden dimension: $d_h = D/H$. For the rest of the paper, we omit the head index $h$ and scaling factor $1/\sqrt{d}$ for simplicity. We denote the query, key and value at some position index $i$ by $\mathbf{k}_i$, $\mathbf{v}_i$, $\mathbf{q}_i \in \mathbb{R}^{1 \times d}$, respectively. In this context, the attention output at the $i^{\text{th}}$ token position $\mathbf{y}_i \in \mathbb{R}^{1 \times d_h}$ corresponds to

$$\mathbf{y}_i = \text{softmax}\left(\mathbf{q}_i \mathbf{K}^\top\right) \mathbf{V}. \tag{3}$$

Due to the nonlinearity and normalization property of the softmax function, $\mathbf{Q}\mathbf{K}^\top$ must be computed to obtain the attention weight, followed by value aggregation through the attention weights, $\mathbf{A}\mathbf{V}$, resulting in $O(N^2)$ complexity with respect to the sequence length $N$ for the self-attention.

### 3.2 ABSTRACTIVE ATTENTIONS AND MULTI-SCALE MULTI-RANGE ATTENTION (MSMRA)

We refer to the family of efficient attention approaches in which the lengths of the attention operands are reduced to $M < N$ by applying an abstraction function $\phi(\cdot)$ as *abstractive attentions*. This approach results in reduced attention complexity while retaining the form of basic attention computation in Eq. 3. Many recent approaches follow this template (Wang et al., 2020; Ma et al., 2021; Dosovitskiy et al., 2020). We focus on cases where the operands of attention, i.e., query, key, and value, are abstracted, noting that there are other possible choices, e.g., abstracting only query and key.

In such cases, the attention mechanism is reduced to

$$\overline{\mathbf{y}}_{i'} = \text{softmax}\left(\overline{\mathbf{q}}_{i'} \overline{\mathbf{K}}^\top\right) \overline{\mathbf{V}}, \tag{4}$$

where $\overline{\mathbf{Q}}$, $\overline{\mathbf{K}}$, and $\overline{\mathbf{V}}$ are abstracted queries, keys, and values, respectively, $\overline{\mathbf{q}}_{i'}$ is $i'^{\text{th}}$ row of $\overline{\mathbf{Q}}$, and $\overline{\mathbf{y}}_{i'}$ is the attention output token with the abstracted query token $\overline{\mathbf{q}}_{i'}$ obtained by

$$\overline{\mathbf{q}}_{i'} = \phi\left(\{\mathbf{q}_{i \in \Omega_{i'}}\}\right). \tag{5}$$

In order to restore the resolution of the output, since the query is abstracted, we define a one-to-many mapping function $\psi(\cdot)$ as

$$\mathbf{y}_{i \in \Omega_{i'}} = \{\psi(\overline{\mathbf{y}}_{i'})\}_i. \tag{6}$$

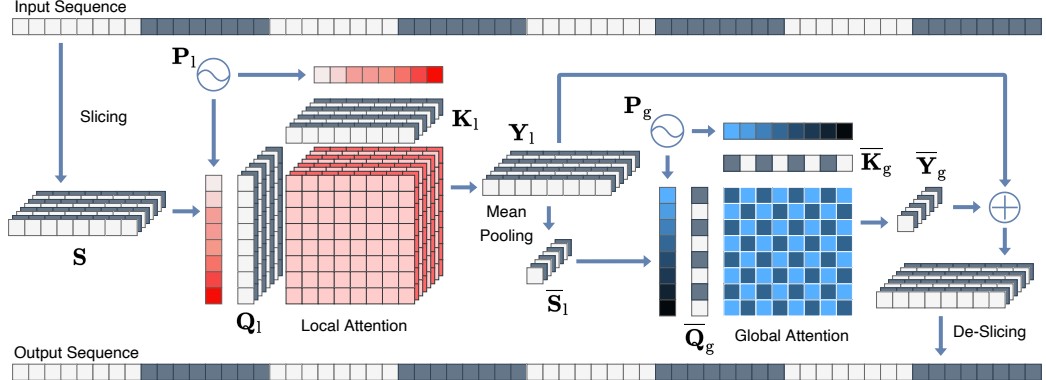

Figure 1: Illustration of Composite Slice Attention. **CSA** consists of a full-resolution local attention with computational complexity $O(NL)$ and a low-resolution global attention with complexity $O(N^2/L^2)$.

A more detailed description of abstractive attention is presented in Section A.1.

Although many previous abstractive attention approaches have achieved sub-quadratic or linear complexities, they typically result in degraded benchmark performance. However, Transformer-based models that leverage multi-scale attention by combining local attention and global attention perform competitively against the standard Transformer. In fact, these models can outperform the standard Transformer on certain tasks, while still maintaining efficiency (Zhu & Soricut, 2021; Zhu et al., 2021; Nguyen et al., 2021; Ren et al., 2021). While other types of MSMRA are described in Section A.2, our proposed attention mechanism is essentially an MSMRA of the form

$$\mathbf{y}_i = \mathbf{y}_{1,i} + \psi\left(\overline{\mathbf{y}}_{\mathrm{g},i'}\right), \tag{7}$$

where $\mathbf{y}_{1,i}$ is the local attention output and $\overline{\mathbf{y}}_{\mathrm{g},i}$ is the global attention output, leveraging a one-dimensional version of the token abstraction used in (Dosovitskiy et al., 2020) for the global attention, but with additional composition of local attention.

## 4 Composite Slice Transformer

We describe the key ideas of Composite Slice Attention (**CSA**) and **CST** network, a Transformer-based model with **CSA** replacing the full softmax dot-product attention in the standard Transformer. **CSA** leverages both full-resolution attention in limited ranges and abstracted attention to capture long-range interactions. Unlike previous approaches, the multi-scale, multi-range attentions are combined through function composition in a serial connection, which allows information passing across representations at different scales and improves expressiveness of the model. See Figure 1 for an illustration of the **CSA** module and Appendix A.3 for full architecture diagrams.

### 4.1 Sequence Representation as Stacked Slices and CSA

In a high-level categorization, the multi-scale multi-range attention of the **CST** corresponds to the combination of block-wise local window attention (Beltagy et al., 2020) and patch-based global attention (Dosovitskiy et al., 2020) in a one-dimensional form. **CSA** layer starts with converting the input sequence $\mathbf{X} \in \mathbb{R}^{N \times D}$ into a stack of slices $\mathbf{S} \in \mathbb{R}^{N/L \times L \times D}$ by slicing it with a fixed length $L$, implemented simply as a Reshape operation. Two attentions with different granularity are then performed sequentially in each direction. First, the batch size $B$ and the number of slices $N/L$ is combined as a new batch size $BN/L$, so that we parallelize the local attention for all slices. Then, the local attention is performed across the tokens within each of these new batches:

$$\mathbf{Y}_1 = \mathrm{softmax}\left(\mathbf{Q}_1 \mathbf{K}_1^\top\right) \mathbf{V}_1, \tag{8}$$

where $\mathbf{Q}_1$, $\mathbf{K}_1$, and $\mathbf{V}_1 \in \mathbb{R}^{BN/L \times L \times d}$ are the queries, keys, and values for each local attention head, computed as $\mathbf{S}\mathbf{W}_{1,q}$, $\mathbf{S}\mathbf{W}_{1,k}$, and $\mathbf{S}\mathbf{W}_{1,v}$, respectively, with $\mathbf{W}_1 \in \mathbb{R}^{D \times d}$. Next, the dimension of length $L$ in the local attention output is collapsed using an abstraction function $\phi$ to get the *slice embedding* $\overline{\mathbf{S}} \in \mathbb{R}^{BN/L \times D}$. Specifically, we use a simple mean pooling $\overline{\mathbf{S}}_s = \phi(\mathbf{Y}_{1,s}) =$

$\sum_{l=0}^{L-1} \mathbf{m}_{s,l} \mathbf{Y}_{s,l} / \sum_{l=0}^{L-1} \mathbf{m}_{s,l}$, where $s$ and $l$ denote the slice index and the token index, respectively, and $\mathbf{m} \in \mathbb{R}^{N/L \times L}$ is the stack of slices of the binary attention mask that is optionally given in the case that the input sequence is padded; e.g., when a data batch has samples with different lengths.[1] Then the second, global attention along the slice dimension is performed to model full-range slice-wise interaction in a reduced resolution:

$$\overline{\mathbf{Y}}_g = \mathrm{softmax}\left(\overline{\mathbf{Q}}_g \overline{\mathbf{K}}_g^\top\right) \overline{\mathbf{V}}_g, \tag{9}$$

where $\overline{\mathbf{Q}}_g$, $\overline{\mathbf{K}}_g$, and $\overline{\mathbf{V}}_g$ is the abstracted queries, keys, and values for the global attention obtained by applying $\mathbf{W}_{g,q}$, $\mathbf{W}_{g,k}$, and $\mathbf{W}_{g,v}$ to $\overline{\mathbf{S}}$. Finally, the output of the global attention is broadcasted to the original resolution and added to the local output:

$$\mathbf{Y}_i = \mathbf{Y}_{l,i} + \mathbf{Y}_{g,i} = \mathbf{Y}_{l,i} + \psi\left(\overline{\mathbf{Y}}_{g,i'},\right) \tag{10}$$

where $\psi(\overline{\mathbf{Y}}_{g,i})$ is a broadcasting function that restores only the sequence length, i.e., $\mathbf{Y}_{g,i} = \overline{\mathbf{Y}}_{g,i'}$ for $i \in \Omega_{i'}$, since the output of the local attention still holds the full resolution. This process can be implemented as a simple broadcast-add operation.

**Slice Extension: Addressing Fine-Grained Context Fragmentation** The local attention of the stacked slice representation of the sequence $\mathbf{S}$ is strictly bounded, resulting in potential context fragmentation (Dai et al., 2019b). Although the global attention models the slice-wise interactions, it may not be sufficient to capture the important fine-grained token dependencies. To allow token-level interaction between slices, we slightly extend each slice into its neighbors, allowing slices to share tokens. This extended local attention can be computed by having keys $\mathbf{K}_{l,\,\mathrm{ext}}$ and values $\mathbf{V}_{l,\,\mathrm{ext}}$ transformed from extended stacked slices $\mathbf{S}_{\mathrm{ext}}$, where $\alpha$ denotes the extension ratio. The extension can be implemented by concatenating shifted-slices in keys and values:

$$\mathbf{S}_{\mathrm{ext}} = \left[\left[\mathbf{0}_{(\alpha-1)L/2}, \mathbf{S}_{:-1,(3-\alpha)L/2:L}\right]_1, \mathbf{S}, \left[\mathbf{S}_{1:,0:(\alpha-1)L/2}, \mathbf{0}_{(\alpha-1)L/2}\right]_1\right]_2 \in \mathbb{R}^{N/L \times \alpha L \times D}, \tag{11}$$

for $\alpha \leq 3$ where the Python notation of indices selection is used.

**Slice Extension and Multi-Scale Causal Mask for Autoregressive Sequence Modeling** In order to apply **CST** to autoregressive sequence processing tasks, we propose custom causal masking and slice extension schemes. For the local token-level attention, we apply an $L \times L$ causal mask map with $-\infty$ above the main diagonal and zero elsewhere to the score tensor $\mathbf{Q}_l \mathbf{K}_l^\top$. However, since the leftmost tokens in each slice have few (and possibly zero) tokens to attend to, we extend the keys and values only on the left-hand side to encourage better fine-grained dependency modeling, i.e.,

$$\mathbf{S}_{\mathrm{AR,\,ext}} = \left[\left[\mathbf{0}_{(\alpha-1)L/2}, \mathbf{S}_{:-1,(3-\alpha)L/2:L}\right]_1, \mathbf{S}\right]_2. \tag{12}$$

For the global attention, an extra care must be taken to prevent leftward information leakage while computing the slice embedding via mean pooling. The diagonal elements in the $N/L \times N/L$ causal mask are set to be $-\infty$ in contrast to the local counterpart. In addition, at the slice index $t$, the shifted query $\mathbf{Q}_{g,t-1}$ is used for query instead of $\mathbf{Q}_{g,t}$ (note that $\mathbf{K}_{g,t}$ is handled by the global causal mask).

**Increased Expressiveness of CST** We mathematically motivate the improved performance of **CST** over competing efficient Transformer models. We show that a given function that **CST** is able to represent is $\epsilon$ away from a rational function with Euclidean degree $4d(\epsilon)$, while competing approaches such as H-Transformer-1D (Zhu & Soricut, 2021) can only approach a rational function with Euclidean degree $d(\epsilon)$. This implies that **CST** is more expressive than other approaches that do not involve compositions of multi-scale attentions. We state the main result below and present the details in Appendix A.5.

**Proposition 1.** *For any fixed $\epsilon > 0$, there exists some Euclidean degree $d(\epsilon) = O(\log(\epsilon))$ such that,*

$$\mathcal{Y}_g^H \subseteq S_{M,d(\epsilon)}^\epsilon, \quad \text{and} \quad \mathcal{Y}_g^{CS} \subseteq S_{M,4\,d(\epsilon)}^\epsilon, \tag{13}$$

*where $M$ corresponds to the total number of weights in* **CST**, *and $S_{M,d}^\epsilon$ is the space of ratios of real analytic functions that are $\epsilon$ away from a rational function with Euclidean degree $d$ with input in $\mathbb{R}^M$.*

---

[1] Normalization with the sum of the binary mask, i.e., the number of nonzero mask values, instead of the slice length $L$, avoids biases in the mean computation induced by masked tokens.

## 4.2 MULTI-SCALE VOLATILE INSTANT POSITIONAL EMBEDDING (MS-VIPE)

Since we reduce the input lengths of both global and local attentions, the full positional embedding of the maximum input sequence length is no longer necessary. For the local attention, we can limit the positional embedding length to the attention range (i.e., the slice length $L$), sharing the embedding across slices. In addition, as we aggregate the tokens from each slice for the global attention, it is more natural to have a separate positional embedding of length $N/L$ at the scale of slice embedding instead of aggregating the full-resolution positional embedding with the same process as the token embedding. The total number of parameters is $((L + N/L)D)$, less than that of a conventional positional embedding $(ND)$.

To this end, **CST** uses two positional embeddings $\mathbf{P}_1 \in \mathbb{R}^{1 \times L \times D}$ and $\mathbf{P}_g \in \mathbb{R}^{N/L \times D}$ applied at different scales, in a fashion similar to that used by (Han et al., 2021), but with a few crucial differences: first, instead of adding the positional embedding to the stacked slices of token embedding at the embedding layer and having aggregated positional information as the layers stack up (Press et al., 2021), **CST** applies them instantly at the corresponding attentions in each layer before the linear transformations. Second, the positional embeddings are applied for the queries and the keys, not for the values, to prevent accumulation of positional information in sequence representations. Since the positional embedding in all layers are added, they can be accumulated over the layers and can undesirably dominate the contextual information at top layers which potentially leads to performance degradation. Our experiments in Section A.6.2 show that the multi-scale volatile instant positional embedding (MS-VIPE) is more effective as compared to the conventional absolute full-length positional embedding. Equations (8) and (9) are rewritten as:

$$\mathbf{Y}_1 = \text{softmax}\left(\{(\mathbf{S} + \mathbf{P}_1)\,\mathbf{W}_{1,q}\}\{(\mathbf{S} + \mathbf{P}_1)\,\mathbf{W}_{1,k}\}^\top\right)\mathbf{S}\mathbf{W}_{1,v}, \tag{14}$$

$$\overline{\mathbf{Y}}_g = \text{softmax}\left(\{(\overline{\mathbf{S}} + \mathbf{P}_g)\,\mathbf{W}_{g,q}\}\{(\overline{\mathbf{S}} + \mathbf{P}_g)\,\mathbf{W}_{g,k}\}^\top\right)\overline{\mathbf{S}}\mathbf{W}_{g,v}. \tag{15}$$

For the extended local attention, we modify the corresponding positional embedding $\mathbf{P}_1$ to have the extended length $\alpha L$, similarly to $\mathbf{K}_{1,\text{ext}}$.

## 4.3 COMPLEXITY AND PARAMETER REDUCTION IN CST

**CST** has *linear plus decimated quadratic* complexity $O(NL + N^2/L^2)$ compared to the $O(N^2)$ complexity of the standard Transformer. However, in a practical range of sequence lengths, e.g., from a few hundreds to a few tens of thousands, **CST** has a comparable or better efficiency than other efficient transformers with linear complexity $O(NM)$ with choices of the abstraction lengths $M$, e.g., from 64 to 256 or higher for even longer sequences, since the slice length $L$ for **CST** is typically less than $M$, even with additional for query, key, and value transformations of $3(N/L)d^2$, which is almost negligible. Furthermore, unlike most efficient transformers that can have similar or higher complexity than the standard transformers with short input sequences, **CST** has better efficiency even in such cases. See Section A.4 for more details of practical complexity analysis.

## 5 EXPERIMENTS

To demonstrate the computational efficiency and sequence modeling capability of **CST**, we evaluate our model in three different contexts: (1) bidirectional long-range dependency modeling on classification tasks (2) word-level auto-regressive language modeling and (3) masked language modeling and transfer learning to natural language understanding on short sequences. Throughout this section, unless stated otherwise, it is implied that we train each model from random initialization, reporting its test performance from the model with the best validation result in each case.

### 5.1 BIDIRECTIONAL LONG-RANGE DEPENDENCY MODELING

**Datasets and Baselines** The Long Range Arena (LRA) benchmark (Tay et al., 2020b) is a suite of classification tasks that evaluate long-range dependency modeling capabilities with datasets from several different modalities with lengths ranging from 1k to 16k. We evaluate **CST** on the five tasks broadly used in literature, *ListOps, Text, Retrieval, Image*, and *Pathfinder*, where the maximum

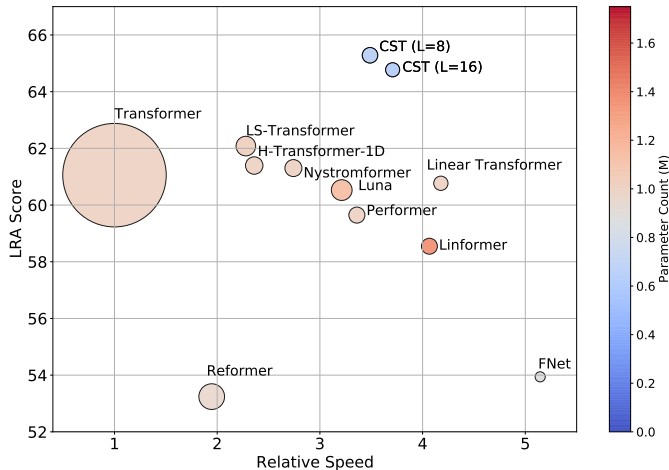

Figure 2: LRA score (y-axis), relative speed (x-axis), relative GPU memory usage (circle radius), and parameter count (color). With competitive efficiency and smaller model sizes, **CST**s outperform strong efficient Transformer baselines as well as the standard Transformer by significant margins. The speed and memory usage measurements can be different depending on devices and implementations.

Table 1: Test accuracy on Long Range Arena (LRA) benchmark.

| Model | ListOps | Text | Retrieval | Image | Pathfinder | Average |
|---|---|---|---|---|---|---|
| Transformer (ours) | **39.92** | 64.20 | 82.11 | 44.25 | 74.79 | 61.05 |
| Transformer (Tay et al., 2020b) | 36.37 | 64.27 | 57.46 | 42.44 | 71.40 | 54.39 |
| Transformer (Xiong et al., 2021) | 37.10 | 65.02 | 79.35 | 38.20 | 74.16 | 58.77 |
| FNet | 36.74 | 64.37 | 74.29 | 44.08 | 50.22 | 53.94 |
| Linear Transformer | 37.40 | 65.16 | 82.73 | 43.81 | 74.74 | 60.77 |
| Performer | 37.25 | 64.65 | 81.69 | 41.73 | 72.91 | 59.65 |
| Cosformer | 38.26 | 64.59 | 82.17 | 41.91 | 67.72 | 59.73 |
| Reformer | 18.04 | 60.94 | 79.19 | 40.38 | 67.66 | 53.24 |
| Linformer | 38.71 | 58.73 | 79.88 | 45.31 | 70.11 | 58.55 |
| Luna | 38.36 | 65.20 | 81.88 | 43.16 | 74.04 | 60.53 |
| Nyströmformer | 37.50 | 65.01 | 80.42 | 51.13 | 69.73 | 60.76 |
| H-Transformer-1D | 38.21 | 66.09 | **84.19** | 46.03 | 72.45 | 61.39 |
| Long Short Transformer | 38.86 | 64.93 | 83.88 | 47.45 | 75.28 | 62.08 |
| Scatterbrain | 38.71 | 63.52 | 81.12 | 41.01 | 70.11 | 58.89 |
| **CST** ($L = 8$) | 37.45 | **77.23** | 84.15 | 51.27 | 76.32 | **65.28** |
| **CST** ($L = 16$) | 37.90 | 73.60 | 84.00 | **51.39** | **76.97** | 64.77 |

sequence length is 4k. We compare **CST**'s efficiency and performance against several state-of-the-art efficient Transformer models. Specifically, we evaluate our model and other Transformer models from a group of non-abstractive attention methods: FNet (Lee-Thorp et al., 2021), Linear Transformer (Katharopoulos et al., 2020), Performer (Choromanski et al., 2020), Cosformer (Qin et al., 2021), a group of abstractive attention methods: Reformer (Kitaev et al., 2019), Linformer (Wang et al., 2020), Luna (Ma et al., 2021), Nyströmformer (Xiong et al., 2021), and another group of multi-scale multi-range attention methods: H-Transformer-1D (Zhu & Soricut, 2021), Long Short Transformer (Zhu et al., 2021), and Scatterbrain (Chen et al., 2021a), as well as the standard Transformer (Vaswani et al., 2017). We outline experimental setup for the Long Range Arena in Appendix A.6.1. Our setup is favorable to the baseline Transformer to provide a fair condition to the challengers, explaining the difference in performances of Transformers from previous work.

**Results** Our experimental results on the LRA benchmark are given in Table 1. We observe that **CST** surpasses the state-of-the-art efficient Transformers with large margins. The best performance is achieved using a slice length 8, outperforming the Transformer baseline by 4.23 points in average score across all tasks. The largest improvements are observed on the *Text, Retrieval*, and *Image* tasks, while a minimal degradation is observed on the *ListOps* task. In addition, as discussed in Section 4.1, **CST** has the added benefit of requiring fewer parameters for a given model size thanks

Table 2: Autoregressive language modeling perplexity on WikiText-103 with $N = 256$. Results for other models are taken from (Nguyen et al., 2021).

| Model | Validation Perplexity | Test Perplexity |
|---|---|---|
| Transformer | 33.15 | 34.29 |
| Linear Transformer | 37.27 | 38.40 |
| FMMFormer (2-kernel + Band$_{20}$) | 35.10 | 36.11 |
| **CST** ($L = 32, \alpha = 3$) | 34.18 | 34.98 |

Table 3: **CST** ablation study on WikiText-103 with varied slice length $L$ and extension ratio $\alpha$ with $N = 256$.

| $L$ | $\alpha$ | Valid PPL | Test PPL | Relative Speedup |
|---|---|---|---|---|
| 8 | 1 | 39.15 | 39.78 | $1.50\times$ |
| 8 | 2 | 36.59 | 37.33 | $1.51\times$ |
| 8 | 3 | 36.10 | 36.78 | $1.49\times$ |
| 16 | 2 | 35.78 | 36.52 | $1.52\times$ |
| 16 | 3 | 35.28 | 36.04 | $1.47\times$ |
| 32 | 2 | 34.83 | 35.60 | $1.54\times$ |
| 32 | 3 | 34.18 | 34.98 | $1.51\times$ |

Table 4: **CST** WikiText-103 test perplexity with $N = 1024$. Results for other models are taken from (Chen et al., 2021a)

| Attention | Test PPL |
|---|---|
| Full Attention | 25.258 |
| Reformer | 27.68 |
| Performer | 66 |
| Scatterbrain | 26.72 |
| **CST**($L = 32, \alpha = 3$) | 26.82 |

to its positional embedding characteristics, although this benefit may be marginal as the network size increases.

We also measure relative speed, and relative GPU memory usage for each model under comparison on a `NVIDIA Tesla V100` GPU with 32GB memory on *Retrieval* dataset with the sequence length of 4k and report the results in Figure 2. Here, relative speed is computed as the number of training steps per second divided by that of Transformer, and GPU memory usage is measured using the PyTorch command, `torch.cuda.max_memory_allocated()`, both on the *Retrieval* task. It is clearly seen that **CST** outperforms all models while having competitive computation and memory efficiency to other efficient Transformer models.

We present an ablation study regarding the choice of positional embedding and aggregation method on the LRA benchmark in Section A.6.2. In summary, variants of **CST** consistently outperform all other efficient models (Table 7), except that switching the MSMRA composition order, i.e. attentions in global-local order, performs slightly worse than Long Short Transformer (Zhu et al., 2021). We demonstrate that the composition of MSMRA in the local-global order and the proposed positional embedding are the most important components contributing to **CST**'s accuracy. While the best overall performance is achieved with $L = 8$, we find that the slice length $L$ has less effect than other hyperparameters. We also find that attention outperforms an MLP-based local token mixing scheme and that mean pooling aggregation outperforms max pooling and linear projection (Table 8).

## 5.2 AUTOREGRESSIVE LANGUAGE MODELING

In this section, to evaluate its applicability as language model, we conduct an autoregressive language modeling experiment on the word-level WikiText-103 dataset (Merity et al., 2016).

**Dataset and Experimental Setup**   The WikiText-103 (Merity et al., 2016) dataset is a collection of featured articles on Wikipedia. It has 103M/218K/246K words in training/validation/test sets, respectively. The task in this experiment is to predict the next word given the history of past words. We follow the experimental setup in (Nguyen et al., 2021) and that in (Chen et al., 2021a) for the context window length 256 and 1024, respectively, and train **CST**s to compare the results with their reported results. While the dataset allows for a larger context, we used the 256 and 1024 context window lengths to match the baselines. **CST** also uses the causal mask and slice embedding described in Section 4.1. More discussion of the experimental setup can be found in Appendix A.7.1.

**Results**   We report the perplexities of the best performing variant of **CST** ($L = 32, \alpha = 3$) compared to other state-of-the-art efficient Transformer models in Table 2 and Table 4. **CST** outperforms other efficient Transformer including Linear Transformer (Katharopoulos et al., 2020) and Performer (Choromanski et al., 2020), kernel method-based linear-complexity models, and

Table 5: Bidirectional MLM and NLU transfer learning evaluation on GLUE benchmark.

| Model | MLM | COLA | MNLI | MRPC | QNLI | QQP | RTE | SST | STSB | GLUE Average |
|---|---|---|---|---|---|---|---|---|---|---|
| BERT$_{base}$ | **4.84** | **60.63** | **83.35** | **91.17** | **90.04** | **87.67** | **69.68** | **92.89** | **87.43** | **82.86** |
| Nyströmformer (64) | 6.77 | 39.39 | 78.75 | 87.23 | 87.35 | 85.64 | 54.87 | 90.48 | 84.28 | 76 |
| Performer (64) | 6.44 | 46.64 | 77.48 | 82.23 | 85.7 | 85.8 | 57.76 | 90.02 | 78.00 | 75.46 |
| Luna (64) | 5.63 | 42.63 | 78.26 | 82.94 | 85.78 | 84.18 | 59.21 | 90.94 | 75.89 | 74.98 |
| FNet | 8.62 | 44.58 | 75.15 | 84.00 | 84.15 | 84.61 | 63.18 | 89.33 | 82.84 | 75.98 |
| **CST** ($L = 16, \alpha = 1$) | 6.00 | 58.49 | 76.45 | 81.76 | 84.44 | 85.24 | 59.57 | 91.74 | 73.77 | 76.43 |
| **CST** ($L = 16, \alpha = 2$) | 5.42 | 58.87 | 78.04 | 82.39 | 85.52 | 85.70 | 60.29 | 92.20 | 75.59 | 77.33 |
| **CST** ($L = 16, \alpha = 3$) | 5.28 | 55.98 | 79.68 | 83.49 | 84.83 | 86.90 | 61.37 | 92.20 | 81.67 | 78.27 |
| **CST** ($L = 32, \alpha = 1$) | 5.77 | 60.34 | 78.47 | 83.06 | 85.17 | 86.36 | 59.93 | 91.74 | 78.26 | 77.92 |
| **CST** ($L = 32, \alpha = 2$) | 5.30 | 57.34 | 80.73 | 85.85 | 86.31 | 87.08 | 59.21 | 91.97 | 85.95 | 79.31 |
| **CST** ($L = 64, \alpha = 1$) | 5.56 | 59.69 | 81.12 | 89.42 | 88.21 | 87.42 | 61.37 | 92.43 | 87.76 | 80.93 |

FMMformer, while being comparable to Scatterbrain (Chen et al., 2021a), noting that the latter two are MSMRA-based models. In Table 3, we provide an ablation study demonstrating the impact of $L$ and $\alpha$ on validation and test perplexity. We observe that longer local attention length leads to better perplexities while being much shorter than the the context window length, e.g. **CST** with $L = 32, \alpha = 3$ has the local window length $64$ that is $4$ and $16$ times smaller than the context window lengths $256$ and $1024$, respectively. We believe that addressing the missing global context, as discussed in A.9, and a better hyperparameter search will improve the perplexity. We also observe consistent 1.5x speed-ups across configurations compared to the standard Transformer in Table 3.

**Additional Experiment on PG-19** We additionally conduct an experiment on PG-19 dataset (Rae et al., 2019). We use various combinations of sequence length $N$, slice length $L$, and extension ratio $\alpha$ to match the attention complexity of **CST**s to those of Transformers with $N = 256$ and $512$. While the validation and test perplexities with discussion can be found in A.7.2, we observe **CST** consistently outperforms the Transformer counterparts with the same attention complexities.

## 5.3 BIDIRECTIONAL LANGUAGE MODELING AND TRANSFER LEARNING

We further evaluate CST on bidirectional language modeling and transfer learning on GLUE benchmark with relatively short input sequences, i.e., $N = 128$. Masked language modeling (MLM) was proposed in (Devlin et al., 2018) as a pretraining method of transformer encoder models and it greatly improves downstream natural language understanding (NLU) performances when fine-tuning the pretrained models. We follow the experimental setup of (Devlin et al., 2018) for conducting both pre-training and fine-tuning including datasets, masking rate, batch size, optimizer settings, and evaluation metrics with a few exceptions. We report the MLM validation perplexities and GLUE scores with accuracy on each task for a variation of $L$ and $\alpha$ in Table 5. Further experimental details, more experimental results, and discussion can be found in A.8. **CST** consistently outperforms other efficient transformers and closes the gap to the baseline BERT model by 0.46 validation perplexity and 2.45 points in GLUE score.

## 6 CONCLUSION

In this paper, we present Composite Slice Transformer (**CST**), an efficient Transformer-based network equipped with composition of multi-scale multi-range attentions. Using stacked slice representation of input sequences, a **CST** layer performs a set of local fine-grained attention and global coarse-grained attention in a sequential manner at a low complexity cost of $O(NL + N^2/L^2)$. In addition to the reduced complexity, we also show that **CST** improves performance on various sequence modeling tasks. On Long Range Arena, word-level autoregressive language modeling on WikiText-103, masked language modeling, and natural language understanding benchmarks, **CST** significantly surpasses strong baselines including recently proposed efficient Transformers, and sometimes the standard Transformer. We highlight limitations and potential directions for future work in A.9.

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

# A APPENDIX

## A.1 ABSTRACTIVE ATTENTIONS

Abstractive attentions are the family of efficient attention approaches in which the lengths of the attention operands are reduced to $M(< N)$ by applying an abstraction function, resulting in reduced complexity of the attention while retaining the form of basic attention computation in Eq. 3. Abstractive attentions can be further categorized to either *resolution preserving* or *non-preserving*, according to which operands are chosen to be abstracted. Resolution non-preserving attention is the result of abstracted queries, further producing abstracted output. This categorization is determined by the requirement of the given task. For example, tasks such as language modeling and machine translation

require the full resolution at the output to be retained. In such cases, it is common to have only abstracted keys and values while the query resolution is retained; the abstractive attention can be expressed as

$$\mathbf{y}_i = \text{softmax}\left(\mathbf{q}_i \overline{\mathbf{K}}^{\top}\right) \overline{\mathbf{V}}, \tag{16}$$

$$\overline{\mathbf{K}} = \left[\overline{\mathbf{k}}_0, ..., \overline{\mathbf{k}}_{j'}, ..., \overline{\mathbf{k}}_{M_k}\right]_1, \tag{17}$$

$$\overline{\mathbf{k}}_{j'} = \phi\left(\{\mathbf{k}_{j \in \Omega_{j'}}\}\right), \tag{18}$$

where $\Omega_{j'}$ denotes the abstraction range with the cardinality $|\Omega_{j'}| = M_k$ for the $j'^{\text{th}}$ key abstraction $\overline{\mathbf{k}}_{j'}$ and $\phi(\cdot) : \mathbf{K}_{\Omega_{j'}} \in \mathbb{R}^{|\Omega_{j'}| \times d_h} \to \overline{\mathbf{k}}_{j'} \in \mathbb{R}^{1 \times d_h}$ is a many-to-one abstraction function. The abstracted values $\overline{\mathbf{V}}_{j'}$ can be expressed similarly. The queries $\mathbf{Q}$ can be similarly abstracted to $\overline{\mathbf{Q}}$ as

$$\overline{\mathbf{q}}_{i'} = \phi_q\left(\{\mathbf{q}_{i \in \Omega_{i'}}\}\right), \tag{19}$$

where $\overline{\mathbf{q}}_{i'}$ is the $i'^{\text{th}}$ row of $\overline{\mathbf{Q}}$. The attention mechanism is reduced to

$$\overline{\mathbf{y}}_{i'} = \text{softmax}\left(\overline{\mathbf{q}}_{i'} \overline{\mathbf{K}}^{\top}\right) \overline{\mathbf{V}}, \tag{20}$$

where an attention output vector $\overline{\mathbf{y}}_{i'}$ is obtained at each abstract position $i'$. In order to restore the resolution of the output, we define a one-to-many mapping function $\psi_y$ as

$$\mathbf{y}_{i \in \Omega_{i'}} = \{\psi_y\left(\overline{\mathbf{y}}_{i'}\right)\}_i. \tag{21}$$

Resolution non-preserving abstraction is often used for the tasks where the full-resolution output is not needed such as sequence-level classification problem. However, in some cases, with additional processing leveraging representations at a lower layer, e.g., cross attention with input tokens, it is possible to restore the resolution (Dai et al., 2020; Jaegle et al., 2021).

## A.2 MULTI-SCALE MULTI-RANGE ATTENTIONS (MSMRA)

Although many previous abstractive attention approaches have achieved sub-quadratic or linear complexities, they typically come at the cost of degraded benchmark performance. However, Transformer-based models that leverage multi-scale attention by combining local attention and global attention perform competitively against the standard Transformer; in fact, these models can outperform the standard Transformer on some benchmarks, while still maintaining efficiency (Zhu & Soricut, 2021; Zhu et al., 2021; Nguyen et al., 2021; Ren et al., 2021).

The local attention, also known as sliding window attention, limits the attention range to the vicinity of query locations. That is, the key abstraction with the whole abstraction range and the query location-dependent abstraction function becomes $\overline{\mathbf{K}}_{\text{l},i} = \phi_{k,i}^{\text{sliding}}(\mathbf{K}) = \mathbf{K} \odot (H(i-j-w/2) - H(i-j+w/2))$ for the $i^{\text{th}}$ query token, where $H$ is the Heaviside step function, $w$ is the window length, and $\odot$ is the element-wise product. This results in the following equation:

$$\mathbf{y}_{\text{l},i} = \text{softmax}\left(\mathbf{q}_i \overline{\mathbf{K}}_{\text{l},i}^{\top}\right) \overline{\mathbf{V}}_{\text{l},i} \tag{22}$$

For better computational efficiency, block-wise key abstraction in Equation equation 22 can be adopted as $\overline{\mathbf{K}}_{\text{l},i} = \phi_{k,i}^{\text{block}}(\mathbf{K}) = \mathbf{K} \odot (H(t_i-j-w/2) - H(t_i-j+w/2))$ for a block-wise attention where $t_i = (b-1/2)w$ for the block index $b$ such that $(b-1)w \le i < bw$.

For the global attention, abstractive attention with either positional abstractions (Zhu & Soricut, 2021; Yang et al., 2021; Ren et al., 2021) or contextual abstractions (Ma et al., 2021; Zhu et al., 2021) can be employed. The former can be loosely seen as having patch embedding in ViT (Dosovitskiy et al., 2020).

MSMRAs can also be categorized according to how the two attentions are combined. While one approach involves concatenating the abstractions of multi-scale keys and values for a single attention operation (Zhu & Soricut, 2021; Zhu et al., 2021; Yang et al., 2021):

$$\mathbf{y}_i = \text{softmax}\left(\mathbf{q}_i \left[\overline{\mathbf{K}}_{\text{l},i}, \overline{\mathbf{K}}_{\text{g}}\right]_1^{\top}\right) \left[\overline{\mathbf{V}}_{\text{l},i}, \overline{\mathbf{V}}_{\text{g}}\right]_1, \tag{23}$$

another separates attentions at different scales and combining the outputs (Han et al., 2021) (possibly with some weighting coefficients):

$$\mathbf{y}_i = \mathbf{y}_{1,i} + \psi_y \left( \mathbf{y}_{\mathrm{g},i} \right). \tag{24}$$

In this case, other non-attentive methods such as kernel method (Nguyen et al., 2021) can also be used for the global attention.

**CST** belongs to the latter of the two approaches, where the local and global attentions are performed separately and their outputs are combined. This is closely related to Transformer-In-Transformer (**TNT**) (Han et al., 2021); however, since **TNT** has a path for the local attention that is independent of the global attention, information exchange between patches is not allowed for the features in the local attention path. Unlike **TNT**, the composition of multi-granular attentions in **CST** enables two-way information passing. This is more suitable for modeling highly non-stationary data, such as natural language text data for example, where the locality assumption does not hold.

### A.3 Architecture of Composite Slice Transformer Network

We present the overall architecture of **CST** alongside the detailed **CSA** architecture in Figure 3. **CST** consists of a fine-grained local attention with computational complexity $O(NL)$ and a coarse-grained global attention with complexity $O(N^2/L^2)$. A shared multi-scale positional embedding, MS-VIPE, is applied to all layers. **CSA** replaces self-attention sublayer in each Transformer block to form a *Composite Slice Transformer* (**CST**) network.

Two sets of $\mathbf{W}_q, \mathbf{W}_k, \mathbf{W}_v$ transformations in **CST** may also affect the parameter count. While TNT (Han et al., 2021) reduces the hidden dimension for local attention to limit the network size increase, we keep the same dimension for representations at different scales and share the weights between local and global attentions, resulting in no increase in parameter count.

### A.4 Consideration on Practical Efficiency and Effectiveness of CST

In this section, we provide an intuition for how and why a multi-scale multi-range attention with $O(NL + N^2/L^2)$ complexity can be a better alternative to the vanilla self-attention than abstraction attentions with linear complexities in terms of both complexity and modeling capability. Improved expressiveness from composition of them is further discussed in Section A.5. A linear complexity can be easily considered to be more efficient than a quadratic complexity. A linear complexity often accompanies another variable; i.e., the abstraction length $M$ in the case of abstractive attention such as (Wang et al., 2020; Ma et al., 2021), resulting in $O(NM)$ complexity. It is obvious that the quadratic complexity $O(N^2)$ is higher than the linear complexity whenever the abstraction length $M$ is smaller than the sequence length, which is mostly true by the definition of *abstraction*. However, when a notion of sequence decimation comes into play, we come to a little different conclusion. Consider an attention on an abstracted sequence of patch embeddings in (Dosovitskiy et al., 2020) or slice embeddings in **CST**. Then the complexity of the abstractive attention becomes a decimated quadratic $O(N^2/L^2)$. In addition to this, if a full-resolution local attention is used as in **CST** or (Han et al., 2021) as a multi-scale multi-range attention, the complexity becomes $O(N^2/L^2 + N/L \cdot L^2) = O(N^2/L^2 + NL)$. We plot the comparison of the linear complexity $O(NM)$ and $O(N^2/L^2 + NL)$ in Figure 4a with several practical choices of the number of abstractions $M$ and the decimation ratio, e.g., slice length, $L$. Here, one can find that a decimated quadratic complexity attention can have better efficiency a linear complexity attention, when the sequence length is less than a few tens of thousands which is considered as a practical range of sequence lengths in many tasks and data types.

Figures 4b shows effective number of tokens for each abstraction for both cases with the same choices of $M$ and $L$. While a decimation, e.g., slicing, retains constant numbers of tokens in each abstraction, the linear complexity attention methods that uses fixed number of abstraction has linearly increasing effective number of tokens per abstraction with respect to the sequence length. Given a fixed hidden dimension, larger number of effective tokens per abstraction requires the abstraction process to compress more information, resulting in loss of potentially important information and negatively affecting the modeling capability.

Since the complexity of **CST** has a decimated quadratic term $N^2/L^2$, the efficiency benefit compared to linear-complexity method keeps closing as the sequence length becomes larger, and eventually

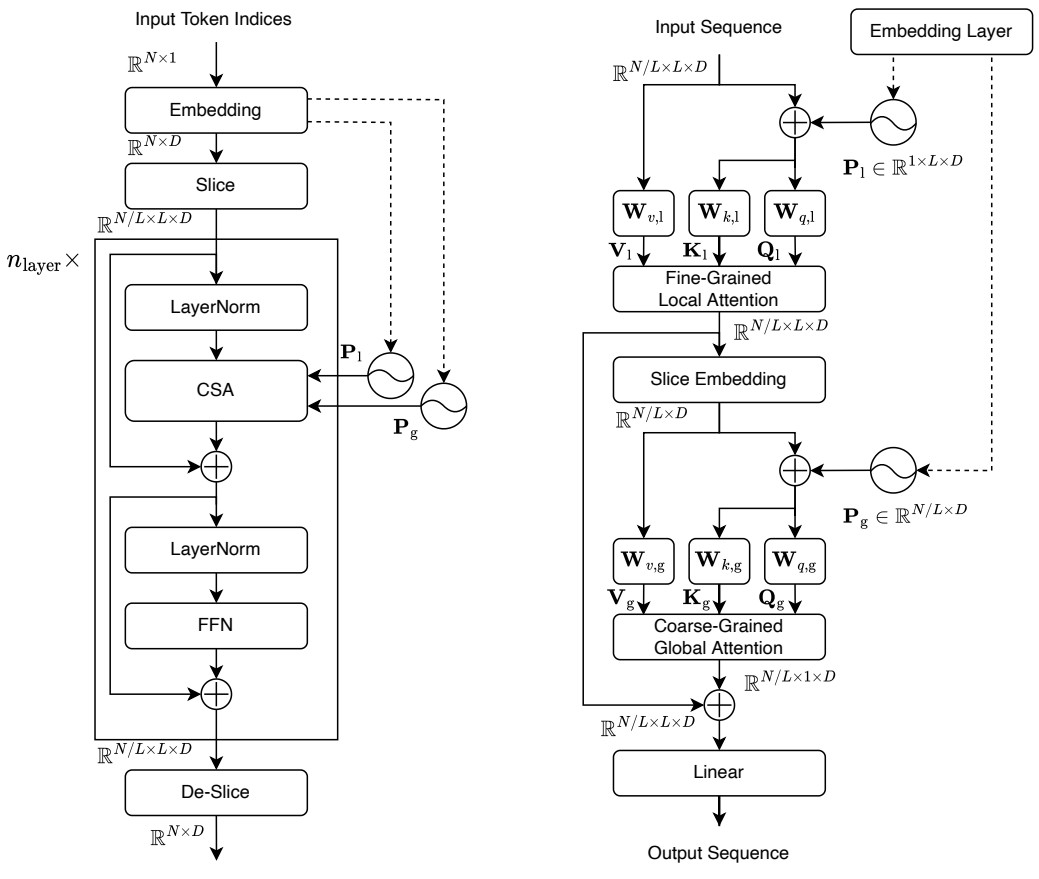

(a) **CST** Network Architecture          (b) **CSA** Architecture

Figure 3: Architectures of **CST** and **CSA**.

the efficiency order is reversed, as it can be seen in 4a. However, we argue that our model can still be beneficial in two aspects. First, it is still more efficient than the standard Transformer. Second, it is possible to find an optimal set of hyperparameters such as the slice length $L$ and the extension ratio $\alpha$ which result in comparably efficient and more effective in modeling capability to the linear-complexity counterpart, considering the discussion made in this section on the effective number of tokens per abstraction.

## A.5 IMPROVED EXPRESSIVENESS OF **CST** SLICE ATTENTION

Recall from Eq.equation 10 that the attention mechanism under consideration takes the form,

$$\mathbf{Y} = \mathbf{Y}_1 + \psi(\mathbf{Y}_\mathrm{g}) \tag{25}$$

where $\mathbf{Y}_1$ represents local attention, $\mathbf{Y}_\mathrm{g}$ represents global attention and $\psi(\cdot)$ is a (linear) one-to-many map. In this section, we will compare **CST** with the H-transformer described in Zhu & Soricut (2021). In particular, we argue that, for a given number of weights (degrees of freedom), our proposed attention mechanism, which involves the *composition* of attention mechanisms, is more expressive than the H-matrix-based approach proposed Zhu & Soricut (2021).Zhu & Soricut (2021) shares similar characteristics with **CST**, but does not involve composition, which explains our superior prediction performance (Section 5).

To explain why this is so, we first define

$$\gamma\left(\mathbf{X};\, \mathbf{W}_q,\, \mathbf{W}_k,\, \mathbf{W}_v\right) = \mathrm{softmax}\left((\mathbf{X}\mathbf{W}_q)(\mathbf{X}\mathbf{W}_k)^T\right)(\mathbf{X}\mathbf{W}_v), \tag{26}$$

which is the standard attention mechanism found in Eq.equation 2. Here, $\mathbf{X}$ is generally a matrix or a 3-tensor; in the latter case, we apply attention independently to each matrix slice along the third

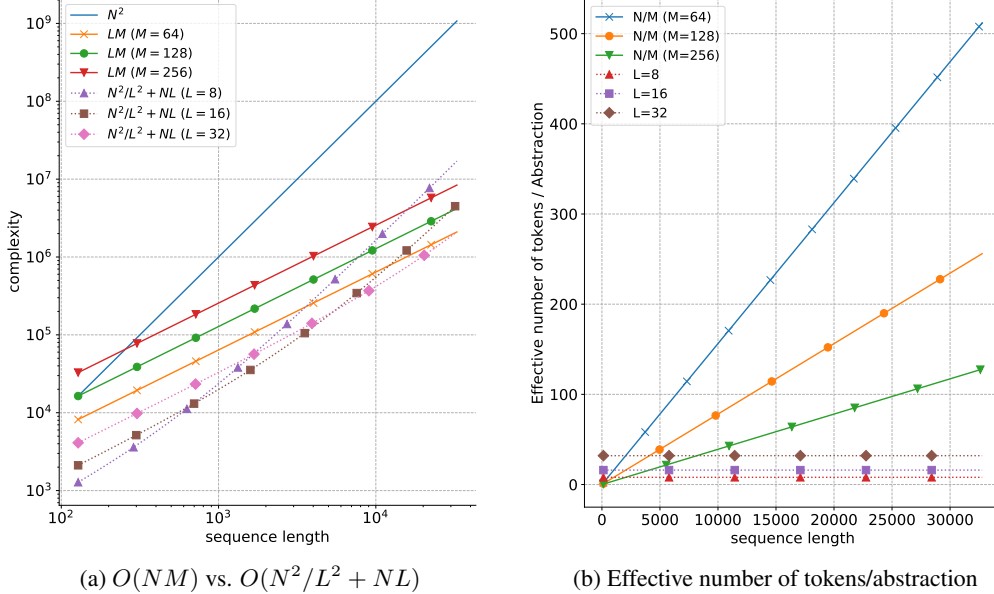

(a) $O(NM)$ vs. $O(N^2/L^2 + NL)$     (b) Effective number of tokens/abstraction

Figure 4: Comparison between linear complexity attention and **CSA**. **CSA** with the complexity $O(N^2/L^2+NL)$ is more efficient than linear complexity methods with $O(NM)$ for practical sequence lengths and choices of model-specific hyperparameters $M$ and $L$. Due to increasing effective number of tokens per abstraction, performance of an abstractive attention with linear complexity can degrade as the sequence length gets large.

dimension. Expressing our local and global attention using this notation leads to,[2]

$$\mathbf{Y}_1^{CS} = \gamma\left(\mathbf{S}; \mathbf{W}_{1,q}, \mathbf{W}_{1,k}, \mathbf{W}_{1,v}\right) \in \mathbb{R}^{N/L \times L \times D} \tag{27}$$

$$\mathbf{Y}_g^{CS} = \gamma\left(\mathbf{Y}_1^{CS} \times_2 \mu; \mathbf{W}_{g,q}, \mathbf{W}_{g,k}, \mathbf{W}_{g,v}\right) \in \mathbb{R}^{N/L \times D} \tag{28}$$

where $\mu = \left[\frac{1}{L}, \frac{1}{L}, ..., \frac{1}{L}\right]^T$ is a fixed vector of mask values, $\mathbf{S}$ corresponds to the stack of $L$-length slices of $\mathbf{X}$ described in Section 4.1, and $\times_2$ indicates a tensor product along the second dimension, i.e.,

$$[\mathbf{Y}_1^{CS} \times_2 \mu]_{ij} = \sum_{k=1}^{L} [\mathbf{Y}_1^{CS}]_{ijk}\, \mu_k \tag{29}$$

which corresponds to mean pooling within each slice. For the purpose of comparison, we focus on a 2-level H-transformer where each element at level 1 has $L$ children at level 0.[3] In this case, it can be shown that the H-tranformer approximation takes the form of Eq. equation 25 with,[4]

$$\mathbf{Y}_1^H = \gamma\left(\mathbf{S}; \mathbf{W}_{1,q}, \mathbf{W}_{1,k}, \mathbf{W}_{1,v}\right) = \mathbf{Y}_1^M \in \mathbb{R}^{N/L \times L \times D} \tag{30}$$

$$\mathbf{Y}_g^H = \gamma\left(\mathbf{S} \times_2 \mu; \mathbf{W}_{g,q}, \mathbf{W}_{g,k}, \mathbf{W}_{g,v}\right) \in \mathbb{R}^{N/L \times D} \tag{31}$$

Note that the main difference between H-Transformer these and **CST** lies in Eq.equation 31 and Eq.equation 28; in the former, the argument to the global attention is $\mathbf{S} \times_2 \mu$, whereas it is $\mathbf{Y}_1^{CS} \times_2 \mu$ in the latter.

---

[2]In practice, it is common to have equal weights at local and global scales, i.e., $W_{l,\cdot} = W_{g,\cdot}$. This is a special case of the analysis presented here and changes the conclusion in no significant way.

[3]More levels have little impact on our conclusion since H-transformers ultimately exhibit the same level of nonlinearity in the weights whatever the number of levels.

[4]Refer to Eq.(29) in Zhu & Soricut (2021) for two levels, i.e., $Y = Y^{(0)} + P^{(0)} \tilde{Y}^{(1)}$. In this case, $P^{(0)}$ corresponds to the one-to-many map $\psi(\cdot)$ whereas $\tilde{Y}^{(1)} = \tilde{A}^{(1)}(R^{(1)^T} V)$ corresponds to applying the level-1 matrix $\tilde{A}^{(1)}$ to the vector $V$ after *averaging over slices* (i.e., applying $R^{(1)^T}$). In this context, this is analogous to applying a global attention to the averaged slices vector, namely, $\mathbf{S} \times_2 \mu$.

To demonstrate our claim, let us first introduce some quantities of interest, starting with the families of parameterized functions that the aforementioned attention mechanisms encompass. First, assume without loss of generality that all the weights are limited to the interval $[-1, 1]$.[5] Then, any attention function learned during the training process must belong to the following sets,

$$\mathcal{Y}_1^{CS} := \left\{ \mathbf{Y} \in \mathbb{R}^{N/L \times L \times D} \ : \ \mathbf{Y}_{ijk} = \gamma_{ijk}(\mathbf{S}; \ \mathbf{W}_{1,q}, \mathbf{W}_{1,k}, \mathbf{W}_{1,v}) \ \forall \ i, j, k \right\} \tag{32}$$

$$\mathcal{Y}_g^{CS} := \left\{ \mathbf{Y} \in \mathbb{R}^{N/L \times D} \ : \ \mathbf{Y}_{ij} = \gamma_{ij}(\sigma(\mathbf{S}) \times_2 \mu \, ; \mathbf{W}_{g,q}, \mathbf{W}_{g,k}, \mathbf{W}_{g,v})), \ \sigma(\cdot) \in \mathcal{Y}_1^M \right\} \tag{33}$$

in the case of **CST** and,

$$\mathcal{Y}_1^H := \left\{ \mathbf{Y} \in \mathbb{R}^{N/L \times L \times D} \ : \ \mathbf{Y}_{ijk} = \gamma_{ijk}(\mathbf{S}; \ \mathbf{W}_{1,q}, \mathbf{W}_{1,k}, \mathbf{W}_{1,v}) \ \forall \ i, j, k \right\} = \mathcal{Y}_l^H \tag{34}$$

$$\mathcal{Y}_g^H := \left\{ \mathbf{Y} \in \mathbb{R}^{N/L \times D} \ : \ \mathbf{Y}_{ij} = \gamma_{ij}(\mathbf{S} \times_2 \mu \, ; \mathbf{W}_{g,q}, \mathbf{W}_{g,k}, \mathbf{W}_{g,v}) \right\} \tag{35}$$

in the case of H-transformer. We want to show that $\mathcal{Y}_g^{CS}$ has more expressive set than $\mathcal{Y}_g^H$, which explains why **CST** exhibits better empirical prediction performance than its competitors. To this end, let us introduce the following quantities:

**Definition 1.** *Let $p(x)$, $q(x)$ be polynomials of Euclidean degree $d$ from $\mathbb{R}^M$ to $\mathbb{R}$. We denote by $S_{M,d}$ the set of rational functions with numerator and denominator of Euclidean degree at most $d$, i.e.,*

$$S_{M,d} := \left\{ \frac{p(\mathbf{x})}{q(\mathbf{x})} \ : \ \deg(p), \deg(q) \le d, \ \mathbf{x} \in \mathbb{R}^M \right\}$$

*imbued with the metric,[6]*

$$d\left( \frac{p(\mathbf{x})}{q(\mathbf{x})}, \frac{a(\mathbf{x})}{b(\mathbf{x})} \right) := ||p(\mathbf{x}) - a(\mathbf{x})||_\infty + ||q(\mathbf{x}) - b(\mathbf{x})||_\infty \tag{36}$$

*where $||f(\mathbf{x})||_\infty = \max_{\mathbf{x} \in [-1,1]^M} |f(\mathbf{x})|$. We also introduce $S_{M,\infty}$ to indicate the space of ratios of real analytic functions, which is a vector space. Further, given any fixed positive $\epsilon \in \mathbb{R}$, we define,*

$$S_{M,d}^\epsilon := \left\{ \frac{p(\mathbf{x})}{q(\mathbf{x})} \in S_{M,\infty} \ : \ d\left( \frac{p(\mathbf{x})}{q(\mathbf{x})}, S_{M,d} \right) \le \epsilon \right\} \tag{37}$$

*as the $\epsilon$-ball surrounding $S_{M,d}$ in the topology induced on $S_{M,\infty}$ by the metric.*

Clearly, for any fixed dimension $M$, $\epsilon > 0$ and Euclidean degrees $d' > d$, the family of functions $S_{M,d'}^\epsilon$ is more expressive than $S_{M,d}^\epsilon$, since the latter is a proper subset of the former, i.e., $S_{M,d} \subset S_{M,d'}$.

Next, we show that the families of functions $\mathcal{Y}_g^{CS}$ and $\mathcal{Y}_g^H$ are in fact subsets of $S_{M,d'}^\epsilon$ and $S_{M,d}^\epsilon$ respectively, for appropriately-chosen values of $\epsilon$, $d'$ and $d$ with $d' > d$. To do so, we need the following result adapted from Trefethen (2017),

**Theorem 1.** *Let $f(\mathbf{x})$ be an analytic function from $\mathbb{R}^M$ to $\mathbb{R}$. Then for every fixed $\epsilon > 0$, there exists a polynomial $p(\mathbf{x})$ of Euclidean degree $d(\epsilon) = O(\log(\epsilon))$ such that,*

$$||f(\mathbf{x}) - p(\mathbf{x})||_\infty = \max_{\mathbf{x} \in [-1,1]^D} |f(\mathbf{x}) - p(\mathbf{x})| \le \epsilon \tag{38}$$

We are now ready to demonstrate our main result,

**Proposition 2.** *For any fixed $\epsilon > 0$, there exists some Euclidean degree $d(\epsilon) = O(\log(\epsilon))$ such that,*

$$\mathcal{Y}_g^H \subseteq S_{M,d(\epsilon)}^\epsilon, \tag{39}$$

*and,*

$$\mathcal{Y}_g^{CS} \subseteq S_{M,4\,d(\epsilon)}^\epsilon, \tag{40}$$

*where $M$ corresponds to the total number of weights.*

---

[5]Any finite bound may be use and has little impact as long as all expressions are subject to the same constraints.

[6]It can be shown that $d(\cdot, \cdot)$ is in fact a proper metric on $S_{M,d}$.

*Proof.* First, recall that (assuming a single head) the parameterized functions $\gamma(\cdot)$ from Eq. equation 26 takes the following form

$$\gamma_{ij}\left(\mathbf{X};\,\mathbf{W}_q,\,\mathbf{W}_k,\,\mathbf{W}_v\right) = \frac{\sum_k e^{[(\mathbf{X}\mathbf{W}_q)(\mathbf{X}\mathbf{W}_k)^T]_{ik}}\left[\mathbf{X}\mathbf{W}_v\right]_{kj}}{\sum_{m,n} e^{[(\mathbf{X}\mathbf{W}_q)(\mathbf{X}\mathbf{W}_k)^T]_{mn}}}. \tag{41}$$

This should be recognized as the ratio of two analytic functions (because the exponential is analytic) paramatrized by the elements of the weight matrices. Following Theorem 1, for every fixed set of weights $\mathbf{W}_g = [\mathbf{W}_q, \mathbf{W}_k, \mathbf{W}_v]$, there exists polynomials $\{p_{ij}^{\mathbf{W}_g}(\mathbf{x})\}$, $\{q_{ij}^{\mathbf{W}_g}(\mathbf{x})\}$ of Euclidean degree $d(\epsilon) = O(\log(\epsilon))$ such that

$$\max_{\mathbf{x}\in[-1,1]^D}\left|\sum_k e^{[(\mathbf{X}\mathbf{W}_q)(\mathbf{X}\mathbf{W}_k)^T]_{ik}}\left[\mathbf{X}\mathbf{W}_v\right]_{kj} - p_{ij}^{\mathbf{W}_g}(\mathbf{x})\right| \le \frac{\epsilon}{2} \tag{42}$$

$$\max_{\mathbf{x}\in[-1,1]^D}\left|\sum_{m,n} e^{[(\mathbf{X}\mathbf{W}_q)(\mathbf{X}\mathbf{W}_k)^T]_{mn}} - q_{ij}^{\mathbf{W}_g}(\mathbf{x})\right| \le \frac{\epsilon}{2}. \tag{43}$$

This means that

$$d\left(\gamma_{ij}\left(\mathbf{X};\,\mathbf{W}_q,\,\mathbf{W}_k,\,\mathbf{W}_v\right),\,\frac{p_{ij}^{\mathbf{W}}(\mathbf{x})}{q_{ij}^{\mathbf{W}}(\mathbf{x})}\right) \le \epsilon \tag{44}$$

and that $\gamma_{ij}(\,\cdot\,;\mathbf{W}_q,\mathbf{W}_k,\mathbf{W}_v) \in S^\epsilon_{M,d(\epsilon)}$. In particular, this shows that $\mathcal{Y}_g^H \subset S^\epsilon_{M_g,d(\epsilon)}$ since every element of $\mathcal{Y}_g^H$ has the form of Eq.equation 41. A similar conclusion can be reached for $\mathcal{Y}_l^H$ and $\mathcal{Y}_l^{CS}$ since elements of these families have the same functional form as those of $\mathcal{Y}_g^H$.

All that remains is $\mathcal{Y}_g^{CS}$. In this case, it suffices to note that the result of polynomial composition creates polynomials with Euclidean degree bounded by the sum of the degree of the polynomials involved in the composition. Indeed, the above analysis shows that elements of $\mathcal{Y}_g^{CS}$ are within $\epsilon$ distance from rational functions of the form,

$$\frac{p_{ij}^{\mathbf{W}_g}\left(\frac{1}{L}\sum_{k=1}^L \frac{p_{ijk}^{\mathbf{W}_l}(\mathbf{x})}{q_{ijk}^{\mathbf{W}_l}(\mathbf{x})}\right)}{q_{ij}^{\mathbf{W}_g}\left(\frac{1}{L}\sum_{k=1}^L \frac{p_{ijk}^{\mathbf{W}_l}(\mathbf{x})}{q_{ijk}^{\mathbf{W}_l}(\mathbf{x})}\right)}, \tag{45}$$

for some polynomials $\{p_{ijk}^{W_l}(\mathbf{x})\}$, $\{q_{ijk}^{W_l}(\mathbf{x})\}$ of degree $d(\epsilon)$. Upon expanding the polynomials and expressing terms using a common denominator, this expression should be recognized as the ratio of two polynomials, each of which has degree $4d(\epsilon)$, following polynomial composition. Thus, we conclude that $\mathcal{Y}_g^{CS} \subset S^\epsilon_{M,4\,d(\epsilon)}$. This proves our claim. $\square$

In other words, Proposition 2 shows that our proposed family of attention mechanisms belongs to $S^\epsilon_{M,4\,d(\epsilon)}$ which possesses more expressive power (since $4\,d(\epsilon) > d(\epsilon)$ ) than the family of functions $S^\epsilon_{M,d(\epsilon)}$ to which global attention mechanism without composition, such as H-transfomers ($\mathcal{Y}_l^H + \psi\left(\mathcal{Y}_g^H\right)$), belong.

We claim that this is the reason why our proposed approach performs better than the approach proposed by (Zhu & Soricut, 2021) and other similar mechanisms where linear combinations of attentions, rather than composition, are used. In other words, for the same number of parameters (weights), our proposed approach can capture more complex attentive interactions than that of (Zhu & Soricut, 2021) and related mechanisms, which leads to a richer set of attentions over which to train the network and, ultimately, better performance.

## A.6 MORE INFORMATION AND DETAILS ON LRA BENCHMARK

### A.6.1 EXPERIMENTAL SETUP

We follow the experimental setup described in (Xiong et al., 2021) with a few exceptions in hyperparameters. Specifically, we use the same Transformer encoder network backbone that consists of 2

Table 6: Hyperparameters for LRA benchmark

| Network Configuration | | | | | | |
|---|---|---|---|---|---|---|
| $D_{\text{embed}}$ | $D_{\text{model}}$ | $D_{\text{FFN}}$ | $d_{\text{h}}$ | $n_{\text{head}}$ | $n_{\text{layer}}$ | $p_{\text{dropout}}$ |
| 256 | 256 | 1024 | 64 | 4 | 2 | 0.1 |

| Task-Specific Training Configuration | | | | | |
|---|---|---|---|---|---|
| Task | $N$ | Batch Size | $\text{LR}_0$ | $n_{\text{step,warmup}}$ | $n_{\text{step,train}}$ |
| *ListOps* | 2k | 32 | $1 \times 10^{-5}$ | 1000 | 20k |
| *Text* | 4k | 32 | $2 \times 10^{-5}$ | 8000 | 20k |
| *Retrieval* | 4k | 32 | $5 \times 10^{-5}$ | 800 | 30k |
| *Image* | 1k | 256 | $2 \times 10^{-4}$ | 175 | 35k |
| *Pathfinder* | 1k | 256 | $1 \times 10^{-4}$ | 312 | 62k |

transformer layers with pre-norm (Xiong et al., 2020) with 4 attention heads and the model dimension 256 across all tasks. Further, we replace the self-attention sub-layer by efficient attention alternatives including the slice attention of **CST**. To obtain label output for classification, we additionally use a classification head network with 2 fully-connected layers of the same hidden dimension with the feed-forward network in the backbone network and a ReLU layer between them. We aggregate the output sequence from the encoder by a mean pooling across the length dimension and feed it to the classification head. For a fair comparison between models with different characteristics, we optimize the learning rates for the baseline Transformer model on each task while keeping all other hyperparameters fixed. For the experimental setup, this allows for a strong baseline model, advantageous for the standard Transformer. We then evaluate each model with a learning rate search within a small bracket $[0.5l_0, l_0, 2l_0]$, where $l_0$ is the base learning rate providing the highest validation accuracy of the Transformer in each task, and report the test accuracy of the model trained with a learning rate that gives the highest validation accuracy among the bracket. We note that our experiments have stronger baselines than those found in literature (Tay et al., 2020b; Xiong et al., 2021) as highlighted in Table 1. For other efficient Transformer models, we perform a model-specific hyperparameter search as described below and report the test accuracy of the best models.

For the Long Range Arena benchmark, we performed experiments on our PyTorch version implementation of the original open-source JAX/Flax code (Tay et al., 2020b) where we followed its dataset preparation procedure and verified the results, and implemented our model on it. Since we found the performances of LRA tasks have relatively high variances, especially on certain tasks such as *ListOps*, we first performed a hyperparameter search with the standard Transformer with the base setup in (Xiong et al., 2021). The hyperparameters we used for the search include number of layers $\{2, 4\}$, number of attention heads $\{2, 4\}$, and model dimension $\{128, 256\}$, and learning rate $\{1, 2, 5\} \times 10^{\{-3, -4, -5, -6\}}$. In addition, for better convergence, we also increased the number of training steps in some tasks. The hyperparameters we determined after the search can be found in Table 6. As this step is essentially a hyperparameter tuning for the baseline Transformer, we note that we have stronger baseline Transformer model than those reported in (Tay et al., 2020b) and (Xiong et al., 2021), and compare them in Table 1. While fixing all other hyperparameters, we performed a learning rate sweep using a learning rate bracket as described in 5.1. For each model and task, we pick the best model on the validation dataset with early stopping, and report its accuracy on the test dataset. For the model training, we used AdamW optimizer (Loshchilov & Hutter, 2017) with linear learning rate warmup and decay, but without weight decay and other training hyperparameters as shown in Table 6.

Since each efficient Transformer has its own hyperparameters and they can affect the performance, we tried a few different values, around the default values used in each paper. Specifically, they are: the number of random features $\{64, 128, 256\}$ in Performer (Choromanski et al., 2020), the projection length $\{64, 128, 256\}$ in Linformer (Wang et al., 2020) and Luna (Ma et al., 2021), the number of landmarks $\{64, 128, 256\}$ in Nyströmformer (Xiong et al., 2021), the number of hashes $\{2, 4\}$ in Reformer (Kitaev et al., 2019), the numerical rank of off-diagonal blocks $\{16, 32, 64, 128\}$ in H-Transformer-1D (Zhu & Soricut, 2021), and the local window segment size $\{8, 16, 32\}$ and

Table 7: Performances of **CST** with various slice lengths $L$, extension ratios $\alpha$, positional embeddings **P**, and combinations of local and global attentions, on LRA benchmark

| $L$ | $\alpha$ | **P** | Local-Global | ListOps | Text | Retrieval | Image | Pathfinder | Average |
|---|---|---|---|---|---|---|---|---|---|
| 8 | 1 | MS-VIPE | Composition | 37.45 | **77.23** | 84.15 | 51.27 | 76.32 | **65.28** |
| 16 | 1 | MS-VIPE | Composition | 37.90 | 73.60 | 84.00 | 51.39 | 76.97 | 64.77 |
| 32 | 1 | MS-VIPE | Composition | 39.01 | 69.24 | 83.92 | **54.21** | 76.73 | 64.62 |
| 64 | 1 | MS-VIPE | Composition | 38.91 | 66.87 | 83.59 | 51.83 | **78.12** | 63.86 |
| 8 | 1 | Conventional | Composition | 38.26 | 69.39 | 83.21 | 47.77 | 70.46 | 61.82 |
| 16 | 1 | Conventional | Composition | **39.31** | 68.22 | 82.68 | 49.02 | 71.30 | 62.11 |
| 32 | 1 | Conventional | Composition | 38.81 | 66.09 | 82.16 | 47.81 | 71.16 | 61.21 |
| 64 | 1 | Conventional | Composition | 37.60 | 65.28 | 82.27 | 47.58 | 72.49 | 61.04 |
| 16 | 2 | MS-VIPE | Composition | 37.90 | 72.81 | **84.62** | 53.41 | 74.87 | 64.72 |
| 16 | 3 | MS-VIPE | Composition | 37.60 | 70.08 | 84.42 | 48.95 | 74.52 | 63.11 |
| 16 | 1 | MS-VIPE | Parallel | 37.85 | 71.15 | 84.02 | 46.05 | 75.13 | 62.84 |
| 16 | 1 | MS-VIPE | Global First | 36.19 | 73.09 | 84.16 | 43.36 | 71.30 | 61.62 |

Table 8: Performances of **CST** with different local modeling and slice aggregation methods, on LRA benchmark. ($L$=16, $\alpha$=1)

| Local Modeling | Aggregation | ListOps | Text | Retrieval | Image | Pathfinder | Average |
|---|---|---|---|---|---|---|---|
| Attention | Mean Pooling | 37.90 | 73.60 | 84.00 | 51.39 | 76.97 | 64.77 |
| MLP ($d_{hidden} = 16$) | Mean Pooling | 37.65 | 66.12 | 81.66 | 45.47 | 74.51 | 61.08 |
| MLP ($d_{hidden} = 32$) | Mean Pooling | 37.25 | 69.70 | 83.63 | 46.79 | 74.19 | 62.31 |
| Attention | Max Pooling | 38.05 | 73.72 | 84.61 | 47.88 | 73.97 | 63.65 |
| Attention | Linear Projection | 37.70 | 73.36 | 82.33 | 50.24 | 72.79 | 63.28 |

the rank of dynamic projection {32, 64} in Long Short Transformer (Zhu et al., 2021). For Long Short Transformer, we optionally used additional 1D depth-wise convolution with the kernel 35 as described in (Zhu et al., 2021). For scatterbrain (Chen et al., 2021a), we follow the default setting for the model specific hyperparameters as in the official code release. Specifically, for text task, we have cluster size for query/key = 16, number of features = 16. For Listops task, we have cluster size for query/key = 32 and number of features = 32. For image and pathfinder task, we have cluster size = 16 and number of features = 16. For retrieval task, we have cluster size for query/key = 64 and number of features = 64. Additionally, we have number of hashes = 2 for all tasks. While each option produces different performances requiring different computation and memory costs, we found variants in each model have similar accuracy and report only the best accuracy among them and the corresponding complexity as comparing them is out of scope of this paper.

### A.6.2 ABLATION STUDY ON LRA

We investigate the effect of each component in **CST** by performing an ablation study. We set the slice length of 16 without extension for local attention as the base configuration. Then, we train a **CST** variant from scratch where one component is changed. Specifically, we consider the variations in: 1) Slice length, 2) Extended local attention, 3) Positional embedding, and 4) Composition of MSMRA. The test accuracies of the variants are given in Table 7. We note that all variants of **CST** consistently outperforms other models, except for one case of the global first composition.

**Slice Length** First, we vary the slice length $L$ and test how the local attention range and abstraction length for the global attention affects the accuracy. Intuitively, one can expect a trade-off between fine-grained token interaction modeling and coarse-grained global context capturing, since in the extreme cases either local attention or global attention converges to the standard self-attention[7]. While each task has its own best $L$, shorter slice length tends to give better overall accuracy. One possible

---

[7]**CSA** converges to the standard self-attention with the global attention at $L = 1$ or the local attention at $L = N$ with $\mathbf{V} = \mathbf{X}\mathbf{W}_v^2$

interpretation is that $L = 8$ or $16$ provides better-formed tokens to the global attention in a dynamic manner by the local attention.

**Local Attention Extension Ratio**   While slight improvements are observed on *Retrieval* and *Image* tasks, on the contrary to our expectation, extended local attention does not always improve the accuracy on LRA tasks. We conjecture it is because LRA datasets has tokens of very fine granularity, i.e., byte-level or pixel-level, and abstraction of the extended local attention output may result in redundant information for global attention. Thus, we evaluate the effect of the extension ratio on a dataset with a coarser-grained tokenization in the Section 5.2.

**Position Embedding**   We compare the MS-VIPE with the conventional learned positional embedding that is applied at the bottom embedding layer of the network. MS-VIPE shows consistent improvement over the conventional positional embedding except for the *ListOps* task where the fined-grained order of each token is essential in this particular task.

**Combination of MSMRA**   Instead of the proposed composition of MSMRA that has a serial connection of MSMRA in the local-global order, we try two different connections, i.e., parallel and global-local connection. Both of them degrade the accuracy compared to the local-global composition, supporting effectiveness of our design.

**Local Modeling with MLP**   As shown in (Tolstikhin et al., 2021) and (Liu et al., 2021a), a multilayer perceptron (MLP) can effectively replace the attention without loss of accuracy in some applications. In **CST**, since the slice length is fixed and relatively short, use of MLP instead of attention can be considered as another source of efficiency with a small increase of parameters. We evaluate a variant of **CST** with 2-layer MLP replacing the local attention, with 2 different hidden dimensions $d_{\text{hidden}} = 16$ or $32$ while $L = 16$ and $\alpha = 1$. MLP in each slice is equivalent to 1D convolution with the kernel size and the stride set to the same with the slice length. We observe that using attention for local modeling module is still preferred in terms of accuracy.

**Slice aggregation**   We further evaluate use of different aggregation methods from the simple mean pooling. Similarly to (Rae et al., 2019), we try max pooling and linear projection. Again, a linear projection is equivalent to 1D convolution with the kernel size and stride set to $L$. While showing comparable results, they lead to degraded performances in Image and Pathfinder tasks. One can expect better accuracy from the linear projection as it is a general form of other pooling methods, but this result can be interpreted as limitation of a simple linear projection for modeling sequence translations. The mean pooling has already generalized well, and with better initialization, linear projection is expected to converge to the mean pooling or slightly better.

## A.7   More Information and Details on Autoregressive Language Modeling

### A.7.1   Experimental Detail of Autoregressive Language Modeling

For the autoregressive language modeling experiment, we use an open-source language modeling experiment framework (Schlag et al., 2021) and plug in our **CST** implementation. We follow all training setup including dataset preparation and hyperparameters given in (Dai et al., 2019a), and evaluate with the small-sized network configuration: 16 layers, 8 attention heads, 128 model dimension, and 2,048 hidden dimension. We train all models using the Adam optimizer (Kingma & Ba, 2014), the cosine annealing learning rate scheduler with 2000 warmup steps while the base learning rate is $2.5 \times 10^{-4}$ for 500K steps with the batch size of 96.

We also conduct autoregressive language modeling experiments with 1024 sequence length. Here, we roughly follow the experiment setup used in (Chen et al., 2021a). The larger model has 512 model dimension instead of 128, and we change the learning rate to $5 \times 10^{-4}$, number of steps to 90K, and batch size to 32, while keeping all the remaining hyperparameters same as that of the small model.

Different choices of slice length $L$ and extension ratio $\alpha$ results in the same local attention range. For instance, the combination of $L = 8, \alpha = 3$ has the same key/value lengths with those in the $L = 16, \alpha = 1$ setting, i.e., $L \times (3-1)/2 = L$. However, since the abstraction length and the resulting granularity of the global attention only depends on $L$, the choices of these hyperparameters can affect

Table 9: Autoregressive language modeling results on PG-19. Models within double-lined sections have matched attention complexities. Models at the top of each section is the baseline Transformer while others are **CST** variants.

| Network | $L$ | $\alpha$ | Context Length $N$ | Valid PPL | Test PPL | Attention Complexity |
|---|---|---|---|---|---|---|
| Transformer | - | - | 256 | 19.66 | 18.70 | $C_1 = O(256^2)$ |
| **CST** | 64 | 3 | 512 | 17.61 | 16.67 | $1.00 \times C_1$ |
| **CST** | 32 | 3 | 1024 | 16.55 | 15.60 | $1.02 \times C_1$ |
| **CST** | 16 | 2 | 2048 | **16.19** | **15.28** | $1.00 \times C_1$ |
| **CST** | 16 | 3 | 2048 | 15.93 | 15.01 | $1.25 \times C_1$ |
| Transformer | - | - | 512 | 17.29 | 16.39 | $C_2 = O(512^2)$ |
| **CST** | 64 | 3 | 2048 | 15.29 | 14.39 | $1.00 \times C_2$ $(0.25 \times C_3)$ |
| **CST** | 32 | 3 | 4096 | **15.11** | **14.15** | $1.06 \times C_2$ $(0.27 \times C_3)$ |
| Transformer | - | - | 1024 | 15.57 | 14.76 | $C_3 = O(1024^2)$ |

the performance differently. We examine how the performance is affected by this configuration. To this end, we evaluate **CST** with combinations of a set of slice lengths $\{8, 16, 32\}$ and a set of extension ratios $\alpha$ $\{1, 2, 3\}$.

To ensure that the current query token has no interaction with the future key tokens, we design the causal mask and local attention extension scheme to better suit **CSA** as described in Section 4.1, and use it throughout this experiment.

### A.7.2 Autoregressive Language Modeling with Matched Attention Complexity to Transformers on PG-19

We additionally perform evaluation of **CST** on PG-19 dataset (Rae et al., 2019) compare to Transformers where their attention complexities are matched, i.e., $O(N^2/N^2 + 0.5(\alpha+1)NL) \simeq O(N^2)$. We use GPT-2 (Radford et al., 2019) implementation in Huggingface transformers library (Wolf et al., 2020), and use the same architecture to implement **CST** equipped with CSA and MS-VIPE. We train all models for 1M training steps using the batch size 32. We use AdamW optimizer with the learning rate$= 1e - 3$, $\beta = [0.9, 0.99]$, and the weight decay$= 0.1$. We report the validation and test perplexities in Table 9. Efficient attention in **CST** enables longer context length with the same complexity in attention computation, leading to significantly improved perplexity compared to the Transformer baseline. Note that, with the current architecture in these cases, the overall complexity can be higher in **CST** since there still are complexity increases in linear layers when they are processing longer inputs. We leave application of caching techniques as future work, that saves complexity in linear layers to match the overall network complexity while enjoying the long-range attention.

### A.8 More Information and Details on Bidirectional Masked Language Modeling and GLUE Benchmark

In this section, we present more details about the experimental setup and results on the bidirectional masked language modeling (MLM) task Devlin et al. (2018) and the natural language understanding tasks in the GLUE benchmark Wang et al. (2018) with discussion.

### A.8.1 Experimental Setup

We pre-train transformer models with the masked language modeling (MLM) objective on Book-Corpus Zhu et al. (2015) and English Wikipedia. After pre-training, the models are fine-tuned on downstream natural language understanding tasks from GLUE benchmark Wang et al. (2018). We follow the experimental setup of Devlin et al. (2018) for conducting both pre-training and fine-tuning experiment, including datasets, masking rate, batch size, and optimizer settings, with several exceptions. We pre-train the models for 900k steps with sequences of 128 tokens. This corresponds to the first phase of BERT pre-training and we consider it to be enough for evaluating on GLUE tasks while expediting the experiment.

With the baseline network architecture of BERT$_{base}$ Devlin et al. (2018), we replace its self-attention layer with different types of efficient attentions such as Nyströmformer, Performer, Luna, and FNet.

For downstream NLU tasks, we also follow the setup of Devlin et al. (2018) with one minor change: instead of prepending [CLS] token in the input sequence, we use mean pooling to get the classifier input vector from the sequence output, as discussed in the following part of this section.

### A.8.2 RESULTS AND DISCUSSION

As shown in Table 5, we can see that **CST** outperforms competing efficient transformers in terms of MLM perplexity. It can be seen that, while **CST**s achieve consistently lower perplexity than most of the efficient transformers, its perplexity decreases as slice length and extension ratio increases. However, **CST** does not outperform BERT$_{base}$ on contrary to our results on the LRA benchmark. We conjecture this is due to fine-grained masking strategy in the original MLM objective. The MLM pre-training stage involves replacing random tokens in the input sequence with a dummy token that the model must predict. After applying the positional embedding, replacing a standard token with a dummy token, i.e., [MASK], translates to injecting high-frequency noise into the input sequence. **CST** and other efficient attention alternatives are approximations to the full attention, which must truncate the high-frequency modes in exchange for faster evaluation. In MLM pre-training, all of the input sequences have high-frequency content that attention approximations have difficulty capturing. Meanwhile, the sequences in the LRA benchmark are long, low-frequency sequences which can be efficiently compressed. An alternative pre-training method could eventually be developed specifically for efficient transformers that circumvents this issue.

On downstream NLU classification tasks, **CST** also outperforms among the efficient transformers. But with under-performing pre-trained models compared to BERT, similar degradation trend is observed in evaluation on downstream tasks after fine-tuning them.

One drawback of the current version of **CST** in the original setting is that the use of [CLS] token is not trivial. [CLS] token can be seen as a global token that summarizes the sequence in the context of the given classification task, and it is often prepended in the input sequence to summarize information from the sequence for the classification purpose. In **CST**, naively prepending it limits attention range in local attention part and also make the summarization in the global attention highly implicit. We believe this can be addressed by an additional well-designed architecture that uses global tokens and input them to classifier. We leave this as a future work and we instead use mean pooling for the experiment for all models.

While the efficiency gain is reduced due to the short sequence length, one can still expect reduction in total FLOPS from using **CST**. For instance, in the simple case of $N = 128$, $L = 64$ and $\alpha = 1$, the complexity of **CST** is $O(128 \cdot 64 + 128^2/64^2)$ which is almost half of that of the standard transformer $O(128^2)$.

### A.9 LIMITATIONS AND FUTURE WORK

### A.9.1 LIMITATIONS

We summarize limitations of **CST** discussed in the previous sections.

As discussed in A.8, there is a mismatch of using token-wise random masking for MLM pretraining. In the abstraction of sequence slices for global attention, high-frequency modes are truncated, i.e., smoothed out, that leads to loss of information. While similar arguments can be made to other transformers with abstractive attention, we plan to carefully analyze it and address the issue by designing an alternative pretraining method.

There are some limitations on the casual mask and slice embedding introduced in Section 4.1, In the global attention, no information of tokens in current slice is taken into account because of the query shift and the attention mask excluding diagonal elements. As a result, there is missing information from coarse-grained modeling in the current design. This is only happening during the training phase, not in the testing, since the next token prediction with sliding window always has access to all tokens in the sequence up to the current time step. Therefore, this mismatch may prohibit the full modeling capability of **CST**. To address the issue, we plan to design more suitable yet efficient

training objective that allows full use of abstractions in the global attention during training. We leave design of more advanced autoregressive modeling as a future work.

**CST** requires additional computations of the complexity $O(N/L)$ for the $Q$, $K$, and $V$ transformation for the global attention, from the local attention output. However, this increase in complexity is negligible compared to the overall computation, especially when $N/L$ is small, as discussed in 4.3.

### A.9.2 FUTURE WORKS

There are several potential avenues for extending **CST**. First, we can easily generalize our work to multi-level slicing with a higher level than two. This will lead to more improvement in expressiveness of the model while adding more degree of freedoms for better efficiency while enabling its application to much longer sequences. Second, as discussed in Section 4.1, more advanced design of autoregressive sequence modeling scheme will further improve the performance. Third, while **CST** is based on fixed-length slicing of the sequence, dynamic or semantic slicing would further improve applicability to non-stationary sequences. With end-to-end training, the model can learn the optimal slicing of the data for given tasks and provide layer-wise dynamic tokens. To this end, efficient realization of the dynamic slicing would be the most challenging task. Finally, as a meaningful next step, modification of **CST** to better fit vision tasks or training a **CST**-based large language model would lead to more general and practical impacts.

