# OpenReview forum: "Composite Slice Transformer: An Efficient Transformer with Composition of Multi-Scale Multi-Range Attentions"
_ICLR.cc/2023/Conference — ICLR 2023 poster_

### Official Review · Reviewer_LFNY · 2022-10-27

**Confidence:** 3
**Correctness:** 3
**Technical Novelty And Significance:** 2
**Empirical Novelty And Significance:** 2
**Recommendation:** 6

**Clarity, Quality, Novelty And Reproducibility:**

Overall the results look promising and the technical details are sufficient. I suggest the authors to better justify the novelty of the proposed method.

**Strength And Weaknesses:**

Strength:
* The authors focus on an important problem in sequence modeling
* The paper is organized well and easy to follow
* The technical details are presented well

Weakness:
* The novelty needs to be better justified
* More ablation study may benefit understanding the proposed method



**Summary Of The Paper:**

The authors propose a hierarchical attention mechanism to model sequence data efficiently. Several experiments have been conducted to demonstrate the effectiveness of the proposed method.

**Summary Of The Review:**

My major concern is on the novelty of the proposed method. It would be better if the authors can further justify how the proposed model is different from MSMRA-based methods, and why the contribution is considered not incremental.

It would be also better if the authors can explore how more slice length choices may affect model performance from the ablation study. The authors use L=8/16 in LRA benchmark and 32/64 in the MLM/NLU transfer learning evaluation, it would be great if the authors can justify why selecting different slice lengths and how this may affect performance here.

Some technical terms also need to be better presented, e.g., the extension rate is not well introduced in the main paper.

---

> ### Author Response · Authors · 2022-11-19
> **Author Response to Reviewer LFNY**
>
> We would like to thank the reviewer for their helpful feedback.
>
> Regarding the novelty of the contribution:
>
> The major differences between our proposed model, CST, from the existing MSMRAs are as follows.
>
> - A key difference is the composition. We propose to convert the input into a two-dimensional tensor and apply a series of attention while aggregation of the first attention (i.e., local attention) output is interleaved for the second attention (i.e., global attention). Then the output of the global attention is broadcasted to the local attention. While the composition of two attention itself contributes to the improved expressiveness of the model as discussed in Section A.5, it further enables fast exchange of globally updated information between fine-grain tokens, thanks to the broadcasting. This mechanism can be seen as the information flow of local-to-local, local-to-global, global-to-global, and global-to-local. Thus, every pair of tokens interact with each other’s globally updated information, in a single layer.
>
> - We further propose a new positional embedding, namely, multi-scale volatile instant positional embedding (MS-VIPE). As we convert the one-dimension sequence input to a multi-scale representation, given that positional embeddings discount the pairwise contextual similarities by their distance, it is natural to weigh the amounts of discounts in local and global attention differently. Volatility and instantness are other important aspects of the proposed positional embedding. They are applied to each layer, instead of applying them to the embedding layer, in the beginning, positional information is fully accounted for in computing self-attention. And, because of the instantness, they are only applied in queries and keys, but not to values, to avoid an accumulation of the positional information across layers. Otherwise, positional information can dominate the token representation prohibiting each token contains enough contextual information.
>
> Regarding clarifying aspects of the model, e.g., extension ratio:
>
> We have clarified the definition of extension ratio in Section 4 as suggested and have reworked the main text to present CSA more clearly by moving important descriptions from the appendix.
>
> Regarding further ablation studies on model choices and hyperparameters:
>
> - We have also moved several ablation studies and discussions from the appendices into the main text to emphasize the impact of slice length on the performance of CST for all benchmarks.
>
> - For the choices of slice lengths and extension ratios in the experiments we present, we performed a coarser screening and a finer ablation to determine the hyperparameters. We basically consider them as tunable hyperparameters for given tasks to determine a trade-off between performance and complexity. Slice length determines the level of granularity of micro and macro interactions of information that is also affected by the granularity of the token data. It means that, when it comes to language modeling tasks, depending on the tokenization either character-level, wordpiece-level, or word-level, benefits from fine-grained attention and global attention can be different, so does the choice of the sequence length. While we haven’t reached a conclusion about the universal rule for the choices with better insight, this can also suggest the use of a multi-scale approach where the level of granularity is further expanded above two. In a similar context, the discussion in a recent paper [1] proposes a way to optimize the (implicit) granularity through end-to-end training.
>
> Reference
>
> [1] Tay, Yi, et al. "Charformer: Fast character transformers via gradient-based subword tokenization." arXiv preprint arXiv:2106.12672 (2021).

---

### Official Review · Reviewer_jNhA · 2022-10-28

**Confidence:** 3
**Correctness:** 3
**Technical Novelty And Significance:** 3
**Empirical Novelty And Significance:** 3
**Recommendation:** 6

**Clarity, Quality, Novelty And Reproducibility:**

The paper is clear and easy to follow. The quality of the paper is good but I am not sure how much novelty in the proposed method as explained in the weakness part. There are details on how to reproduce the results.

**Strength And Weaknesses:**

Strength
1. The paper is well written and easy to follow.
2. Although local+global attention has been used in previous works to reduce computational complexity, the author proposed a new form.
3. The proposed method achieves better performance than previous efficient transformer.

Weakness
1. The proposed CSA is similar to previous axial attention [1,2], where they conduct self-attention sequentially along one dimension and then another. The difference may be that the proposed method also used attention abstraction and multi-scale position embedding. It would be better if the author could give some more detailed discussions about the difference from previous works.
2. I am not sure I understand the multi-scale position embedding. First, I am not sure why it is called multi-scale, is it just because the length of local and global sequences are different? Second, how is this different from previous relative positional embedding? And there should be an ablation study to show its effect.

**Summary Of The Paper:**

This paper proposed an effecient self-attention block for transformer structure. The proposed block first splits the input sequence into multiple local sequences. Then it first conducts self-attention on the local sequences and then aggregates local sequences and conducts self-attention on the aggregated sequences to achieve global attention. The proposed method is evaluated on multiple datasets and shows better performance than previous efficient transformer.

**Summary Of The Review:**

I think the paper is well written and the proposed method has some new technical contributions. But I am not sure how significant these contributions are and I would like to see the response from the author.

---

> ### Author Response · Authors · 2022-11-19
> **Author Response to Reviewer jNhA**
>
> We would like to thank the reviewer for their helpful feedback.
>
> Regarding the multi-scale positional embedding:
>
> - The proposed positional embedding, namely, multi-scale volatile instant positional embedding (MS-VIPE), is applied to multiple scales in the sense that one-dimension data (such as text or audio) is converted to a multi-scale representation to have a fine granularity (original resolution) and a coarse granularity (reduced resolution), and then two positional embeddings are added in each granularity. Given that positional embeddings discount the pairwise contextual similarities by their distance, it is natural to weigh the amounts of discounts in local and global attentions differently.
>
> - Volatility and instantness are other important aspects of the proposed positional embedding. Instantness: They are applied to each layer, instead of applying them to the embedding layer in the beginning, positional information is fully accounted in computing self-attention. Volatility: In addition, because they are added in each layer, we only apply them to queries and keys, but not to values, to avoid accumulation of the positional information across layers. Otherwise, positional information can dominate the token representation prohibiting each token contains enough contextual information.
>
> - We have included an ablation study in Table 7 emphasizing the impact of the proposed multi-scale positional embedding. MS-VIPE is one of the essential components in CST for improved performance while being slightly parameter efficient compared to conventional positional embeddings.
>
> Regarding the similarity between CSA and axial attention:
>
> CSA is similar to axial attention [1, 2] in some sense, but also has several major differences that we try to describe to the best of our knowledge.
>
> 1. The choice of positional embedding, which we have discussed above.
>
> 2. Axial attention starts with multi-dimensional data (such as image) as its input. Two consecutive attentions are essentially local attentions in different axis, although it eventually allows global interactions. On the other hand, a basic architecture of CST is based on one-dimensional sequence data (such as text) and reshapes the input sequence into a N/L x L tensor. CST then collapses the slice dimension into N/L global tokens (slice embeddings), which makes the second attention a global one, at a coarse scale than the previous local attention.
>
> 3. CST enables a faster full-duplex information flow since it has the slice aggregation step and a skip connection with a broadcast-add operation. After a local attention followed by a global attention (global interaction exchange happens here), which contains an updated information in aggregated tokens (i.e., slice embeddings), is distributed to the fine-grained tokens. This mechanism makes the information flows local-to-local, local-to-global, global-to-global, and global-to-local, and thus every pair of tokens interact with each other’s globally updated information, in a single layer. On the other hand, axial attention implicitly enables the global interaction between fine-grained tokens (or pixels) via two intersections of the axes. For example, in a 4x4 image, information exchange between the pixel in (1, 1) and the pixel (4, 4) occurs through (4, 1) and (1, 4). However, globally updated information from, for instance, (2, 2) can only be handed to those pixels after propagating 2 or more layers.
>
> 4. Global information in the axial attention can be noisier because of the fine-grained context in each token, compared to the aggregated slice embedding in CST, even when a simple mean pooling is used. When it comes to a long-range information delivery, this aggregation can make the global information more refined and make the information delivery more effective.
>
> 5. Although CST is designed considering one-dimensional data, it can be extended to two-or-more-dimensional inputs, which is one of our ongoing works. For two-dimensional data, a global token (or slice embedding) can be more effective in the form of a patch embedding in ViT [3]. In this case, CST is more related to Transformer-in-Transformer [4] where composition of local and global attention is the key difference, as well as the volatile instant positional embedding.
>
> References
>
> [1] Ho, Jonathan, et al. "Axial attention in multidimensional transformers." arXiv preprint arXiv:1912.12180 (2019).
>
> [2] Huang, Zilong, et al. "Ccnet: Criss-cross attention for semantic segmentation." Proceedings of the IEEE/CVF international conference on computer vision. 2019.
>
> [3] Dosovitskiy, Alexey, et al. "An image is worth 16x16 words: Transformers for image recognition at scale." arXiv preprint arXiv:2010.11929 (2020).
>
> [4] Han, Kai, et al. "Transformer in transformer." Advances in Neural Information Processing Systems 34 (2021): 15908-15919.

---

### Official Review · Reviewer_z3gP · 2022-10-31

**Confidence:** 4
**Correctness:** 4
**Technical Novelty And Significance:** 3
**Empirical Novelty And Significance:** 3
**Recommendation:** 8

**Clarity, Quality, Novelty And Reproducibility:**

The paper is quite clear and very well written. The only problem I see is that the extension ratio (\alpha) is only defined in the appendix. You should consider moving it to the main part of the paper because having it in the appendix makes Table 2 and 3 very difficult to understand.

In terms of novelty, the combination of local and global attention is not novel.
However, the way it is done in this paper is novel (as far as I know) and it seems very simple and intuitive.

In terms of reproducibility, the model and the hyperparameters are clearly defined so it should be quite easy to reproduce the results.
However, I did not see any mention of the intention to publicly release the model’s code. This could lead to a lot more usage from the community.


**Strength And Weaknesses:**

Strengths:
- The model description (besides the extension ratio) is quite clear and well written.
- The proposed way to combine the local and global attention is quite simple and intuitive.
- The authors perform experiments on a wide set of tasks, which show the model’s advantages and wide applicability.

Weaknesses:
- Despite performing experiments on a wide set of tasks, the lengths of the inputs used for auto-regressive and masked language  are always quite small (256 and 1024 for autoregressive LM and 128 for MLM). As the complexity problems of the vanilla transformer are more problematic for long inputs, I do not understand why the authors chose these lengths for the experiments.
- The autoregressive LM experiments are only performed on Wikitext-103. In my opinion the paper would benefit from having experiments on a dataset with more long-range dependencies, such as PG-19 (https://github.com/deepmind/pg19), using a bigger input length. This would allow the reader to understand how good the global attention mechanism is.
- I also think that it would be interesting to have an experiment in which the computational requirements of CST equal the ones of the vanilla transformer (for example by having longer sequences for CST than for the vanilla transformer)  to understand the gains obtained while having the same computational requirements.
- There is no reference to the S4 model (Efficiently Modeling Long Sequences with Structured State Spaces, Gu et al., CLR 2022), which according to the S4 paper results is far better than the CST on the Long Range Arena tasks.


**Summary Of The Paper:**

In this paper the authors propose the Composite Slice Transformer (CST) which consists of a composition of attentions applied to a stacked, slice representation of the input sequence at different scales, coupled with a multi-scale volatile instant positional embedding.

To do this, first the input sequence of length N is divided into N/L slices of size L. Then, local self-attention is performed within each slice of text, to obtain a local context vector for each input token.
These context vectors also go through a mean-pooling layer (one per slice) to obtain a vector that represents each slice. Then global self-attention is performed using the vectors outputted by the pooling layer. Finally, the final context vector that corresponds to each input token is obtained by summing the local context vector with the global context vector (broadcasted with the broadcast-add operation, to restore the sequence length).

As CST has two attention mechanisms with different granularities, it uses two positional embedding layers, one for the local attention and another for the global attention. These embeddings are only summed to the queries and keys, to prevent the accumulation of the positional information over the layers of the model.

For a fixed slice length L, CST has a complexity of O(NL + N^2/L^2) while the vanilla transformer has a complexity of O(N^2), where N is the input sequence length.

The authors perform experiments in three tasks: Long Range Arena, auto-regressive language modeling, and masked language modeling, and compare the results with a broad set of efficient transformers and the vanilla transformer.

On the Long Range Arena, CST leads to the best average score while being the fastest model and one of the efficient transformers that requires less GPU memory.

On auto-regressive language modeling, experiments on the Wikitext-103 with a sequence length of 256 show gains in perplexity when comparing with the Linear Transformer and the FMMformer while having 1.5x speed-up when compared to the vanilla transformer.
Further experiments, using a sequence length of 1024, show improvements over Reformer, Performer, and Scatterbrain.

On masked language modeling with input sequences of 128 tokens, CST leads to a lower perplexity and better GLUE average score than Nystromformer, Performer, Luna, and FNet.


**Summary Of The Review:**

To sum up, I really liked reading this paper and found the model proposed very interesting and widely applicable (as shown by the wide set of experiments in which the model performs well).

However, I found the experiment's setting to be a bit disappointing since the authors only considered small sequence lengths for the language modeling tasks.

---

> ### Author Response · Authors · 2022-11-19
> **Author Response to Reviewer z3gP**
>
> We would like to thank the reviewer for their useful feedback on our work and appreciate their positive impression of our work.
>
> Regarding the short sequence length for language modeling tasks:
>
> We agree with the reviewer’s comment on sequence length choices and admit the absence of evaluation results on language modeling with long sequences.  While we added experimental results on PG-19 language modeling as described in the following rebuttal point, we will try to explain the reason for these choices of sequence lengths.
>
> 1. While we considered evaluation with longer sequences in the LRA benchmark, we tried to follow the experimental setup from previous models for language modeling to compare with their reported performance directly. For example, WikiText-103 with N=256 from FMMFormer[1], N=1024 from Scatterbrain[2], and MLM/GLUE from BERT[3].
>
> 2. While this work focuses mainly on addressing the quadratic complexity bottleneck that arises in long sequences, we believe that it is still valuable to evaluate a model on short sequence tasks, especially for use cases in low-resource devices. Furthermore, unlike the linear-complexity transformers, CST also achieves better efficiency for short sequence lengths that can lead the model to be adopted in a wider range of applications - we have added a discussion in Section 4.3.
>
> Regarding autoregressive language modeling on PG-19:
>
> We gratefully took the reviewer’s suggestion and conducted an additional experiment of language modeling on the PG-19 dataset. We added the experimental setup and results in Section A.7.2 and summarized it as a paragraph in Section 5.2. Our observation is as follows: (Please see Table 9 on Page 23, We haven't included the table here due to the limitation of the number of characters)
>
> Efficient attention in CST enables longer context length with the same complexity in attention computation, leading to significantly improved perplexity compared to the Transformer baseline. Please note that, however, with the current architecture, the overall complexity can be higher in CST since there still is a linear growth of complexity in fully connected layers when they are processing longer inputs. We leave the application of caching techniques to intermediate representations at each layer as future work, which can save this complexity increase in these layers to match the overall network complexity to Transformers while enjoying the long-range attention.
>
> Regarding the lack of reference to S4:
>
> We also have noticed that several recent models such as S4[4], GSS[5], MEGA[6], and SGConv[7] achieve significantly better performances on the LRA benchmark. We have added these models in the related work with citations.
> While it might be out of scope from the reviewer’s comment, our conjecture about their superiority on LRA is as follows: LRA was designed for a quick performance/efficiency comparison of efficient transformers with relatively small-sized datasets and does not require a pre-training step. Meanwhile, Transformers (including efficient Transformers) is a family of networks with some of the highest degrees of freedom and flexibility, and thus usually require a large amount of data to train them. The small-sized experimental setup such as LRA can be highly favorable to models with more structure and strong inductive biases like S4, GSS, MEGA, and SGConv. We also think that those models show a promising research direction for efficient alternatives of self-attention in sequence modeling.
> We believe that these works do not overshadow CST outright, as CST and other efficient Transformer architectures contain more flexibility with fewer inductive biases and will prove useful in applications.
>
> Regarding the remaining comments:
>
> We have moved the definition of the extension ratio from the appendix was moved to the main body.
> We also wished and planned to release the source code, at least for the model implementation. But unfortunately, our organization does not encourage open-sourcing codes. We will continue trying to make it open-sourced. Meanwhile, as we think the model is clearly described in the paper with details including operations and tensor dimensions, we believe a reader with moderate knowledge and experience in deep learning would be able to easily implement it.
>
> References (with only their titles due to limitations of the number of characters)
>
> [1] "Fmmformer: Efficient and flexible transformer via decomposed near-field and far-field attention.".
>
> [2] "Scatterbrain: Unifying Sparse and Low-rank Attention Approximation.".
>
> [3] "Bert: Pre-training of deep bidirectional transformers for language understanding.".
>
> [4] "Efficiently modeling long sequences with structured state spaces.".
>
> [5] "Long range language modeling via gated state spaces.".
>
> [6] "Mega: Moving Average Equipped Gated Attention.".
>
> [7] "What Makes Convolutional Models Great on Long Sequence Modeling?.".

---

### Official Review · Reviewer_i8cg · 2022-11-03

**Confidence:** 5
**Correctness:** 3
**Technical Novelty And Significance:** 3
**Empirical Novelty And Significance:** 3
**Recommendation:** 5

**Clarity, Quality, Novelty And Reproducibility:**

The originality of the method is good, but the presentation is a little bit unclear as many important contents are put in Appendix.

**Strength And Weaknesses:**

[ Strength ]
+ The general idea is new. Existing studies on reducing the quadratic complexity focus on either the global or the local pattern, and their combination is seldom explored. The proposed local-to-global method can have benefits from both patterns, i.e., local for higher efficiency and global for higher accuracy. Therefore, the idea is new and technically sound.
+ Theoretical analysis is solid. The authors give very detailed theoretical analysis on the motivation of the proposed method, which makes the theory solid and rigorous.
+ The performance on several benchmarks is consistently state-of-the-art.

[ Weakness ]
- The main body of the paper is not self-contained. Unfortunately, many important statements and analysis are put into Appendix, e.g., the formulation of CSA at the beginning of Section 4. Without these, readers can hardly understand the content clearly. It is not realistic for all readers to refer to Appendix frequently.
- Some important ablations are not in the main paper. Again, critical ablations such as the speed with respect to the sequence length and the impact of primary hyper-parameters are all put in Appendix, making the paper not self-contained. I strongly advise the authors to re-organize the paper structure and move some important ablations in the experimental section. For example, the two figures in Figure 1 indicate the same meaning, so only retaining one of them could save more space for text.
- Lack of the comparison to some SOTA linear Transformers, e.g.,
[1] Flowformer: Linearizing Transformers with Conservation Flows
[2] COSFORMER : RETHINKING SOFTMAX IN ATTENTION

**Summary Of The Paper:**

This paper proposes a new module that reduces the quadratic complexity of standard Transformers. The key idea is to slice the input sequence into various groups and then perform local and global attention respectively. Experimental results indicate that the local-to-global approach can indeed accelerate the processing speed and save memory costs. Also, the proposed model achieves state-of-the-art performance on several popular NLP benchmarks.

**Summary Of The Review:**

I acknowledge the originality and effectiveness of the proposed method, but the paper presentation is not self-contained and should be carefully re-organized.

---

> ### Author Response · Authors · 2022-11-19
> **Author Response to Reviewer i8cg**
>
> We would like to thank the reviewer for their useful feedback on our work. We appreciate the positive comments about the novelty, performance, and analysis of CST.
>
> Regarding the paper’s organization:
>
> We have worked hard to restructure the paper according to your feedback. We have moved all of the contents regarding the formulation of CSA into the main text, including how we address fine-grain context fragmentation, modifications for autoregressive sequence modeling, and a brief discussion of the theory supporting CST’s performance gains. We have also moved the ablation studies for bidirectional language modeling and autoregressive language modeling into the main text, along with a short discussion. We were unfortunately not able to fit the LRA benchmark ablation study regarding the positional embedding and have left it in the appendix.
>
> Regarding the missing comparison to state-of-the-art efficient Transformer models:
>
> We appreciate the reviewer for directing us to the state-of-the-art efficient transformer models, FlowFormer[1] and CosFormer[2]. During the rebuttal period, we were able to integrate the CosFormer in our LRA benchmark code, thanks to its available implementation taken from https://github.com/OpenNLPLab/cosFormer, and observed CST outperforms it. We have included the result in Table 1 and added a citation and discussion accordingly. We were unfortunately not able to do the same for FlowFormer, but we expect it to perform better than CosFormer with about a 1 point margin according to [2], but still worse than CST, at least, on the LRA benchmark.
>
> [1] Huang, Zhaoyang, et al. "FlowFormer: A Transformer Architecture for Optical Flow." arXiv preprint arXiv:2203.16194 (2022).
>
> [2] Qin, Zhen, et al. "cosFormer: Rethinking Softmax in Attention." arXiv preprint arXiv:2202.08791 (2022).

---

### Author Response · Authors · 2022-11-19
**Authors' response to All Reviewers**

We would like to thank the reviewers for their constructive feedback on our work. According to the feedback, we were able to make a better-looking revised paper. The major changes we made are as follows.

- We have worked hard to restructure the paper according to the reviewer’s feedback and make it self-contained with minimal redirection to appendices. Specifically, we have moved the formulation of CSA into the main text (Section 4), including how we address fine-grain context fragmentation, modifications for autoregressive sequence modeling, and a brief discussion of the theory supporting CST’s performance gains (Section 4.1), while keeping more details in the appendices. We also moved the ablation study results to the main text to highlight the impact of each component of CST on its performance (Section 5).

- We performed additional experiments and added the results with discussion, further demonstrating the effectiveness of CST architecture. The additional experiments include a comparison to another state-of-the-art model, CosFormer, on the LRA benchmark (Section 5.1, Table 1), and language modeling benchmark on the PG-19 dataset with longer sequence lengths while matching computation requirements with Transformer (Section 5.2, Section A.7.2, Table 9).

- We tried to further highlight the advancements and novelty in CST from existing models, especially MSMRA-base models, in each section as well as in the response to each reviewer.

With the changes in revision and the responses that we made to the reviewers, we hope every question and concern on our work is well addressed.

---

### Decision · Program_Chairs · 2023-01-20

**Decision:**

Accept: poster

**Justification For Why Not Higher Score:**

Novelty is a bit limited in my opinion for this paper to deserve a spotlight or oral presentation.

**Justification For Why Not Lower Score:**

The paper still provides valuable contributions and I would like to see it accepted.

**Metareview: Summary, Strengths And Weaknesses:**

This paper proposes a new local-global attention mechanism (composite-slice attention) to overcome the quadratic complexity of standard transformers. The key idea is to slice the input sequence into various groups and then perform local and global attention respectively. Experimental results indicate that the local-to-global approach can indeed accelerate the processing speed and save memory costs, achieving good performance on several popular NLP benchmarks. Overall, there was some disagreement among reviewers about this paper, with most reviewers feeling overall positive. The authors seem to have addressed most of the reviewers' concerns in their rebuttal, adding experiments in a more challenging dataset which benefits more from long context (PG-19) to the revised version of their paper. Therefore I am recommending acceptance.

**Note From Pc:**

if the above contains the word "oral" or "spotlight" please see: "oral" presentation means -> notable-top-5% and "spotlight" means -> notable-top-25%. As stated in our emails, we are disassociating presentation type from AC recommendations